# Transcriptomic profiling of shed cells enables spatial mapping of cellular turnover in human organs

Tal Barkai [1,2], Oran Yakubovsky [1,2,3], Yael Korem Kohanim[4], Keren Bahar Halpern[1], Sapir Shir [1], Noa Oren[1], Michal Fine[1], Paz Kelmer [1,2], Amit Talmon [2,3], Alon Israeli[2,3], Niv Pencovich[2,3], Ron Pery[2,3], Ido Nachmany[2,3] & Shalev Itzkovitz [1✉]

## Abstract

Single-cell atlases provide valuable insights into gene expression states but lack information on cellular dynamics. Understanding cell turnover rates—the time between a cell's birth and death—can shed light on stemness potential and susceptibility to damage. However, measuring turnover rates in human organs has been a significant challenge. In this study, we integrate transcriptomic data from both tissue and shed cells to assign turnover scores to individual cells, leveraging their expression profiles in spatially resolved expression atlases. By performing RNA sequencing on shed cells from the upper gastrointestinal tract, collected via nasogastric tubes, we infer turnover rates in the human esophagus, stomach, and small intestine. In addition, we analyze colonic fecal washes to map turnover patterns in the human large intestine. Our findings reveal a subset of short-lived, interferon-stimulated colonocytes within a distinct pro-inflammatory microenvironment. Our approach introduces a dynamic dimension to single-cell atlases, offering broad applicability across different organs and diseases.

**Keywords** Shed Cells; Cell Turnover; Spatial Transcriptomics; Single Cell Atlas; Gastrointestinal Tract
**Subject Category** Chromatin, Transcription & Genomics

See also: C Kilian & L Adlung

## Introduction

Cellular turnover, defined as the time between the birth and death of a cell, is a fundamental cellular feature that spans several orders of magnitude (Sender and Milo, 2021; Reddien, 2024). Quantifying cellular turnover has important implications in physiology and pathology. Identifying low turnover cells can highlight tissue stem cells, whereas identifying high turnover cells could expose cellular fragilities and unique tissue microenvironments that may be associated with cell death. Quantifying turnover has traditionally required pulse-chase methods, in which specific sub-populations are labeled (Leblond and Walker, 1956), either through DNA-integrating reagents (Bonhoeffer et al, 2000) or, more recently via cellular barcodes (Urbanus et al, 2023; Sankaran et al, 2022). The time that has elapsed until labeling disappeared yielded estimates of cellular lifetime. These approaches entail genetic manipulation and repeated tissue sampling (rather than snapshots as in regular biopsy acquisitions), and are therefore less relevant to healthy humans (Lipkin et al, 1963). Computational tools have been developed to infer cellular dynamics based on single-cell RNAseq (scRNAseq) data through trajectory inferences (La Manno et al, 2018; Trapnell et al, 2014; Setty et al, 2019; Weiler et al, 2024), however, they do not directly inform on cellular turnover rates.

When cells die, they are often shed into the bloodstream or into the outer cavities of tissues, either whole or as cellular fragments. While shed cells can experience changes in gene expression due to apoptosis or other death-related processes (Ngo et al, 2022), they are believed to retain the identity of the tissue and cell type of origin. This principle has been fundamental in the field of liquid biopsies, whereby cellular fragments in blood report on cellular death throughout the body (Mandel and Metais, 1948; Vorperian et al, 2022, 2024; Sadeh et al, 2021; Larson et al, 2021; Chalasani et al, 2023). In the digestive tract, which includes the esophagus, stomach, small and large intestine, as well as the hepato-pancreatic-biliary tree, cells are predominantly shed into the luminal cavities (Bullen et al, 2006). We have recently demonstrated, in both mouse and human, that shed cells profiled transcriptomically in fecal washes reflect cellular states of the gastrointestinal (GI) tract (Bahar Halpern et al, 2023; Dan et al, 2023; Ungar et al, 2022). Here, we combine shed-cell transcriptomics with single-cell and spatial transcriptomics to assign turnover scores to individual cells in spatially resolved expression atlases. We apply this approach to the upper and lower GI tract. We characterize the turnover rates of diverse cell types and identify tissue microenvironments that may promote cell death.

## Results

### Reconstruction of spatially resolved turnover maps

To enable assessment of cellular turnover, we analyze the bulk RNA expression signatures of both shed cells and tissue cells (Fig. 1).

[1]Department of Molecular Cell Biology, Weizmann Institute of Science, Rehovot, Israel. [2]Department of Surgery and Transplantation, Sheba Medical Center, Ramat-Gan, Israel. [3]Gray Faculty of Medical & Health Sciences, Tel Aviv University, Tel Aviv, Israel. [4]Department of Immunobiology, Yale University School of Medicine, New Haven, CT, USA. ✉E-mail: shalev.itzkovitz@weizmann.ac.il

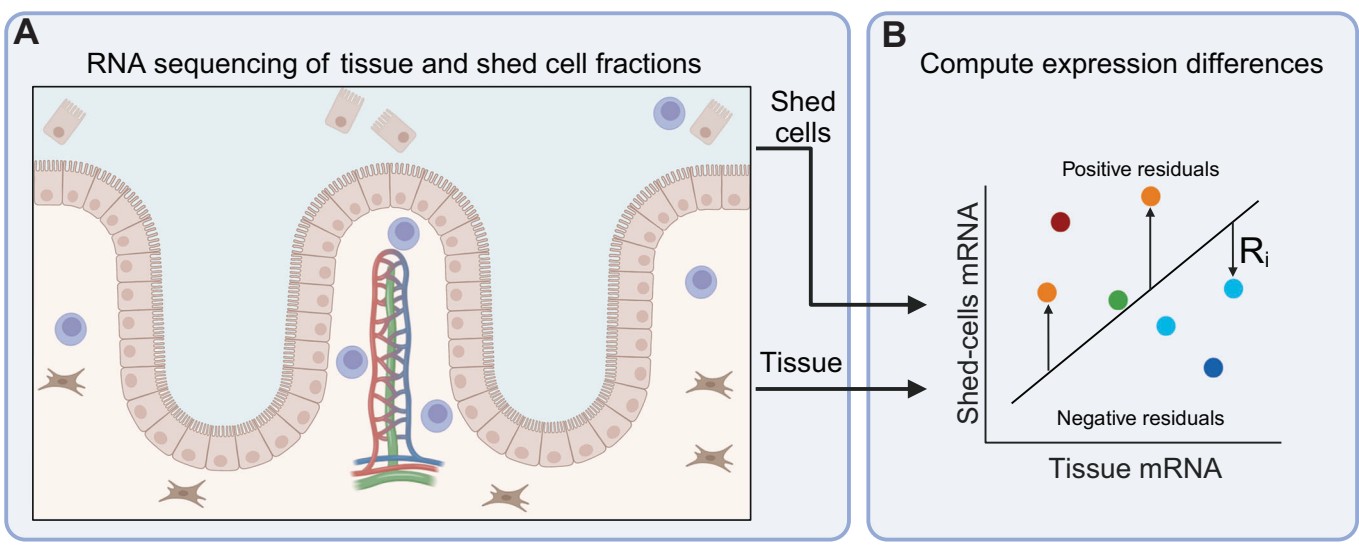

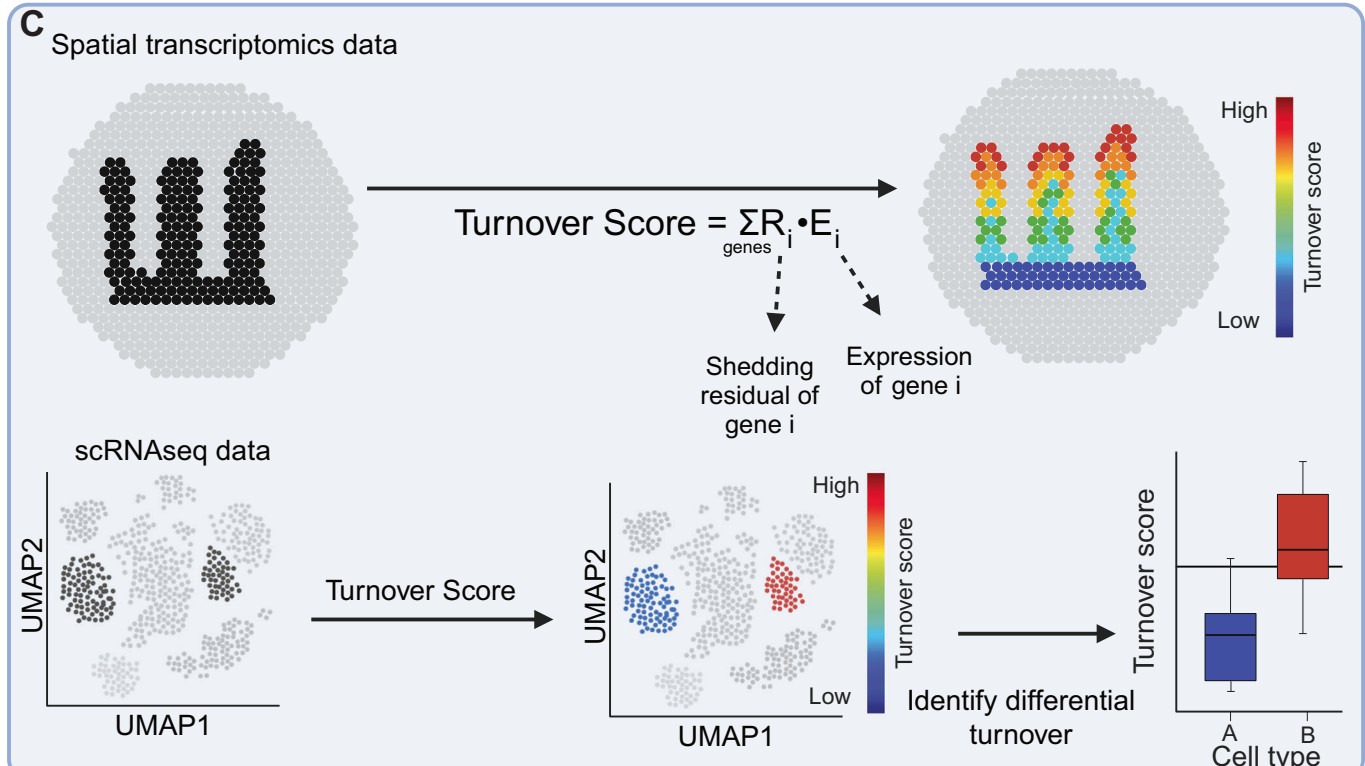

**Figure 1. Strategy for reconstructing turnover maps.**

(**A**) RNA extracted and sequenced from tissues and from the fraction of shed cells. (**B**) Each gene is assigned a residual value measuring the extent to which it is over-/underexpressed in the shed cell fraction compared to the tissue. (**C**) Spots (in spatial transcriptomics data) or cells (in scRNAseq data) are scored according to their turnover score—the sum of gene residual values weighted by expression. This facilitates exploration of turnover patterns of specific cell types or spatial zones and markers of cells with high/low turnover scores.

In the GI tract, where most shedding normally occurs into the luminal cavities, shed cell fractions can be obtained from endoscopic or gastric washes, whereas tissue cells can be obtained from biopsies (Fig. 1A). We regress the expression of the shed cell and tissue fractions and assign to each gene $i$ a residual $R_i$, reflecting the degree to which it is over- or underexpressed in the shed cell fraction compared to the tissue (Fig. 1B).

$$Turnover\ score = \sum_{i=0}^{genes} Residual_i \cdot Expression_i \qquad (1)$$

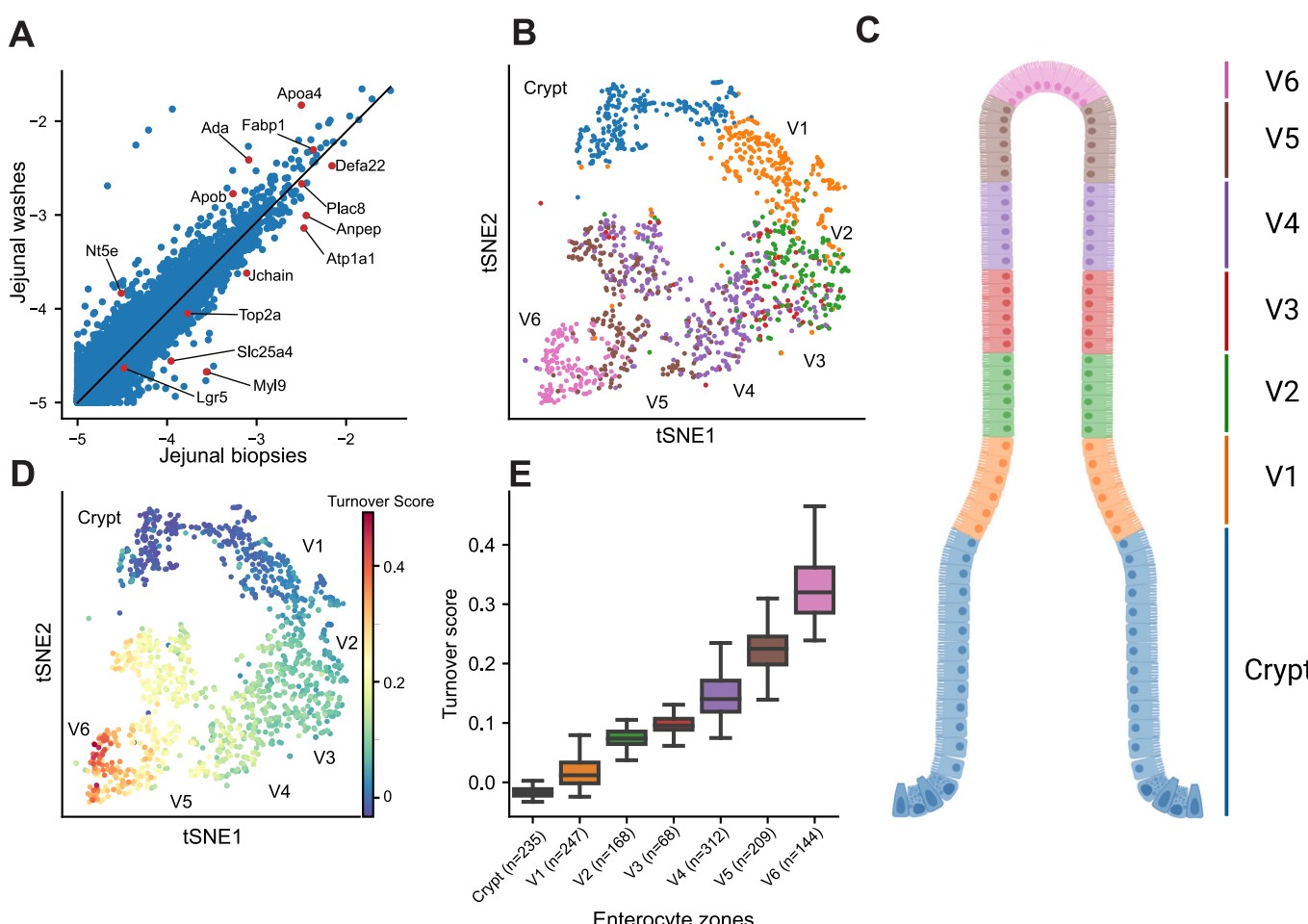

**Figure 2. Turnover patterns in the mouse small intestine.**

(A) Scatter plot of gene expression in mouse small intestinal tissue (biopsies, *x* axis) and washes (*y* axis). Data from Bahar Halpern et al (Bahar Halpern et al, 2023). Each dot is a gene, representative genes expressed more highly in washes or tissue are shown in red. Expression in units of log10(sum-normalized Unique Molecular Identifier (UMI) counts). (B, C) Enterocyte genes shown over a tSNE plot of single cells (B), from Moor et al) and schematically across the crypt–villus axis. Villus zones marked from base (V1) to tip (V6). (B) Each dot is a cell. (D) tSNE map colored by the computed turnover score. Each dot is a cell. (E) Turnover score strongly increases toward the villus tip. Box plots show the median as the center line, boxes span the 25th–75th percentiles, whiskers extend up to 1.5 IQR, outliers are not shown. Turnover scores computed using Eq. (1).

We next analyze single-cell RNA sequencing atlases or spatially resolved expression atlases and compute a turnover score, defined as the sum of gene residuals, weighted by their expression (Fig. 1C, Eq. (1), where Residual$_i$ is the residual of gene$_i$ and Expression$_i$ is the sum-normalized expression of gene$_i$). Tissue cells with high turnover scores are expected to show elevated levels of genes that are more highly expressed in the shed cell fractions (and consequently have positive residuals, Fig. 1C). The ability to combine scRNAseq and spatially resolved atlases with the turnover score enables interrogating turnover patterns of distinct cellular subtypes and tissue zones (Fig. 1C).

To examine our approach, we first focused on the mouse small intestine, an organ in which cellular turnover is well understood (Beumer and Clevers, 2021) (Fig. 2). We used transcriptomics data of tissues and of jejunal washes that consisted of shed cells from Bahar Halpern et al (Bahar Halpern et al, 2023) to extract the genes' residuals (Fig. 2A). Expression of jejunal washes and tissues were

highly correlated (*R* = 0.88, *P* < 1e-323), yet distinct genes showed either positive or negative residuals. While genes with low expression residuals were often stromal genes (e.g., the myofibroblast cell marker *Myl9* and the plasma cell marker *Jchain*, Fig. 2A), epithelial genes showed highly variable residuals as well. Villus tip markers (Moor et al, 2018), such as *Ada* and *Apoa4* had positive residuals, whereas crypt (*Defa22*, *Top2a*) and lower villus markers (Moor et al, 2018) (*Anpep*, *Atp1a1*) had zero or negative residuals (Fig. 2A). We next used scRNAseq data from Moor et al, (Moor et al, 2018), which was annotated according to crypt–villus zones based on landmark genes (Bahar Halpern et al, 2023; Moor et al, 2018) (Fig. 2B,C) to assign turnover scores to individual cells (Fig. 2D), and analyzed turnover scores of distinct zonal enterocyte sub-populations (Fig. 2E). We found that turnover score strongly increases from the crypt towards the tip of the villus, as expected based on the measured shedding patterns in the mouse gut (Bahar Halpern et al, 2023; Cheng and Leblond, 1974).

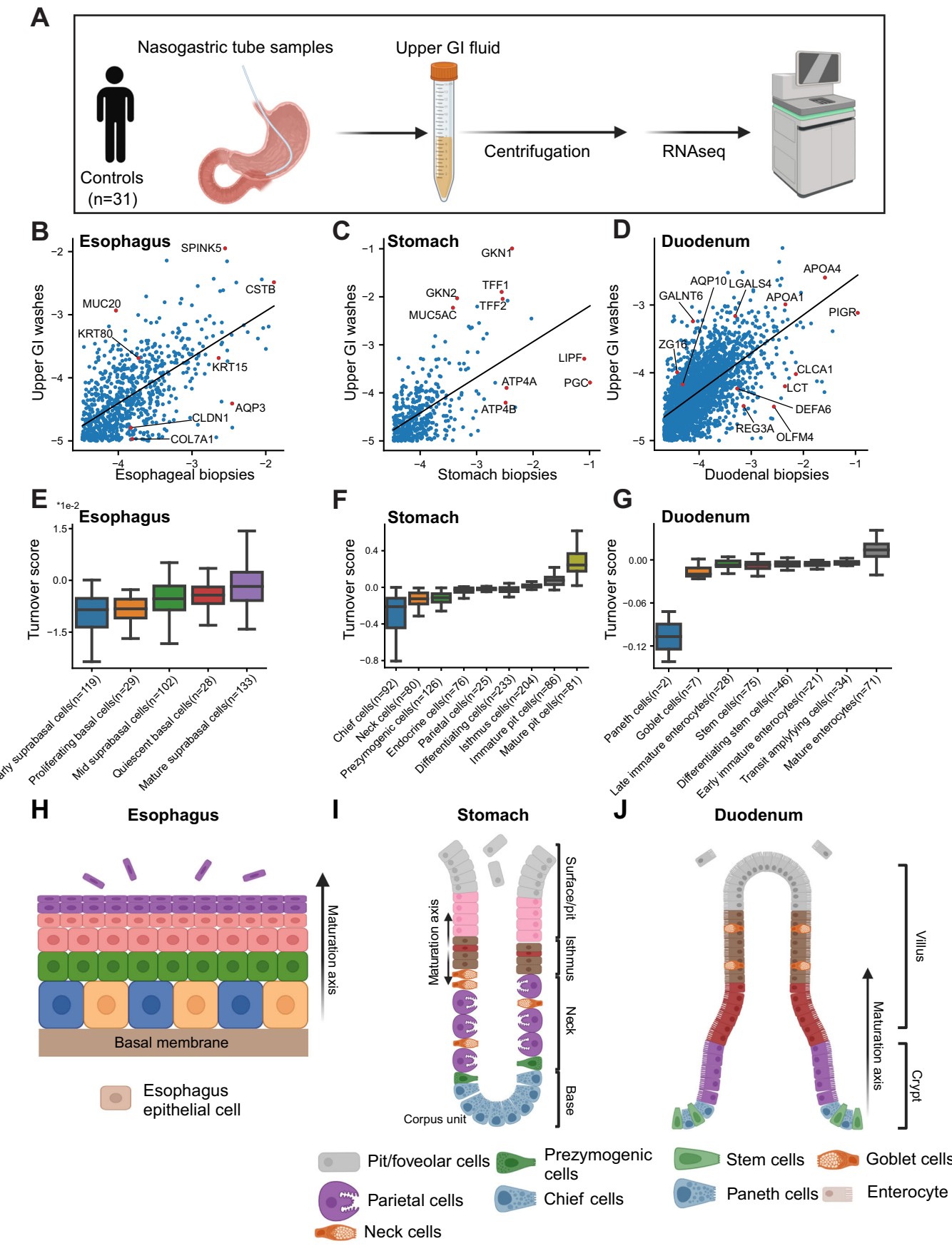

◄ **Figure 3. Turnover maps of the human upper GI tract.**

(A) Acquisition of upper GI fluids (NGT) for transcriptomic profiling of shed cell fractions. (B–D) Scatter plot of gene expression in the NGT and in specific organs of the upper human GI tract. In each panel, each dot is a gene, data include internally normalized expression over the set of genes that are specific to the respective organs—esophagus (B), stomach (C), and duodenum (D). Representative genes expressed more highly in washes or tissue are shown in red. (E–G) Turnover scores of specific sub-populations, annotated by Busslinger et al (Busslinger et al, 2021) ("Methods", external datasets). Box plots show the median as the center line, boxes span the 25th–75th percentiles, whiskers extend up to 1.5 IQR, outliers are not shown. (H–J) Diagram of the zonal locations of the cell types shown in (E–G). Turnover scores computed using Eq. (1).

To validate the zonal trend in enterocyte turnover, we further examined the turnover scores of a recently generated combined single-cell atlas of tissue and shed cells in the mouse intestine (Bahar Halpern, Fig. EV1A,B). These data demonstrated that single-shed enterocytes were similar to villus tip enterocytes both in their general transcriptome, as well as in their turnover scores (Fig. EV1C). Since shed cells may exhibit death-related changes in gene expression upon shedding, we repeated our shedding score calculation while excluding apoptotic genes ("Methods"). We found that the zonal trends in turnover scores remained unchanged (Fig. EV1C,D). We further compared turnover of other mouse jejunal lineages and found that enterocytes and goblet cells showed high turnover scores, whereas immune cells and tuft cells showed relatively lower turnover scores (Fig. EV1E), consistent with their anatomical distance from the lumen.

## Turnover maps of the human upper GI tract

We next turned to analyze cellular turnover patterns in the human upper GI tract. To characterize the shed cell fractions, we assembled a cohort of 31 patients with a mean age of 58.5 ± 16 years, in which 38.7% were female, and the mean body mass index 25.7 ± 4.7 kg/m². To profile the expression of shed cells, we enlisted patients who were hospitalized for surgeries unrelated to upper GI tract malignancies (Fig. 3A). These patients typically receive nasogastric tubes (NGTs) during their procedures, which facilitate the aspiration of stomach fluids. These fluids contain a blend of cells shed from the esophagus, stomach, and even the duodenum. To enrich for the presence of duodenal cells, we aspirated fluids containing bile, noticeable by their green coloration. We extracted RNA from these NGT fluids and applied mcSCRBseq, a sensitive UMI-based RNA sequencing protocol (Bagnoli et al, 2018) (Datasets EV1 and EV2). Resampling the same patients at two different time points, up to 15 min, showed that our method is reproducible within this time frame (Fig. EV2A).

To calculate turnover scores for the upper GI tract organs, we used bulk RNA sequencing of biopsies of the upper GI tract, from GTEX (Lonsdale et al, 2013) and Abadie et al, (Abadie et al, 2020), to identify genes distinctly expressed by the esophagus (834 genes, Dataset EV3, Fig. EV2B), stomach (427 genes, Dataset EV3; Fig. EV2C), and duodenum (2504 genes, Dataset EV3; Fig. EV2D). For each organ, we used only these marker genes for normalization and regression analysis to extract turnover residuals, by comparing the expression in the NGT fluid to bulk RNA sequencing of the respective human organ extracted from GTEX and Abadie et al (Fig. 3B–D). We used these residuals to assign a turnover score to each single cell from the scRNAseq atlas of Busslinger et al (Busslinger et al, 2021).

The esophagus is composed of a stratified epithelium, where stem cells are thought to reside in the most basal layers. We found that turnover score closely follows the anatomical organization of this organ, with low levels in all populations except in the upper-most mature suprabasal cells (Fig. 3E,H). The stomach is composed of repeating anatomical glands termed pits, harboring distinct cell types (Fig. 3I). These include the mucus-secreting pit cells located at the uppermost parts, acid-secreting parietal cells, enriched in the neck, and proteolytic enzyme-secreting chief cells at the base of the gland. As in the esophagus, we found that turnover score follows anatomical location within the gland, with maximal turnover scores for the mature pit cells and lowest scores for chief cells (Fig. 3F). The exceedingly low turnover scores of chief cells correlate with their measured lifetime of half a year in mice (Karam and Leblond, 1993). In the duodenum (Fig. 3G,J), we found that Paneth cells had the lowest turnover score, in line with their reported elongated lifetime of 57 days (Ireland et al, 2005), followed by intestinal stem cells. As in the esophagus and stomach, turnover closely correlated with anatomical location, with the highest turnover for mature enterocytes, residing in the villus upper zones. We also performed the turnover score analysis on Visium data from Harnik et al (Harnik et al, 2024), again demonstrating the clear correlation between the anatomical location of cells along the villus axis and their turnover score (Fig. EV3). This supports the notion that, as in mice, enterocytes in the healthy human small intestine are predominantly shed from the tips of the villi in healthy individuals.

## Spatial turnover maps of the human colon

The colon is composed of repeating anatomical crypts, harboring stem cells at their bottoms. We used shed cell and tissue data from Dan et al, (Dan et al, 2023) (Fig. 4A). This data included bulk RNAseq measurements of colonic biopsies and fecal washes obtained during screening colonoscopies of healthy individuals. We found that genes expressed in the mesenchymal layers of the tissue, such as the muscle cell marker *ACTA2* and the sub-mucosa marker *PLVAP* were depleted in the washes, consistent with the lower luminal shedding of cells residing in these layers. Among the epithelial genes, we found that colonic washes exhibited higher expression of markers of the inter-crypt epithelium (e.g., *AQP8*, Fig. 4A,B), and lower expression of markers of crypt epithelium (e.g., *SLC26A2*, Fig. 4A,B). We used the residuals to assign turnover scores to 8-μm pixels transcriptomically profiled using Visium HD (Fig. 4C,D, Oliveira et al, 2025). We found that the shedding score was significantly elevated in the inter-crypt epithelium compared to all other tissue domains (Fig. 4E, $P < 1e\text{-}323$). In addition, we found that overall colonic representation is more prominent in biopsy and wash compared to serum cfRNA (Fig. EV4A,B) and that biopsy-wash residuals are anti-correlated with biopsy-serum residuals (Fig. EV4C). Examination of colonic-specific gene expression in both serum cfRNA and wash transcriptomics shows that mature colonocytes and goblet cells have more representation in the luminal wash compared to the serum (Fig. EV4D). Luminal cell shedding in the human colon therefore seems to be concentrated in the inter-crypt epithelium.

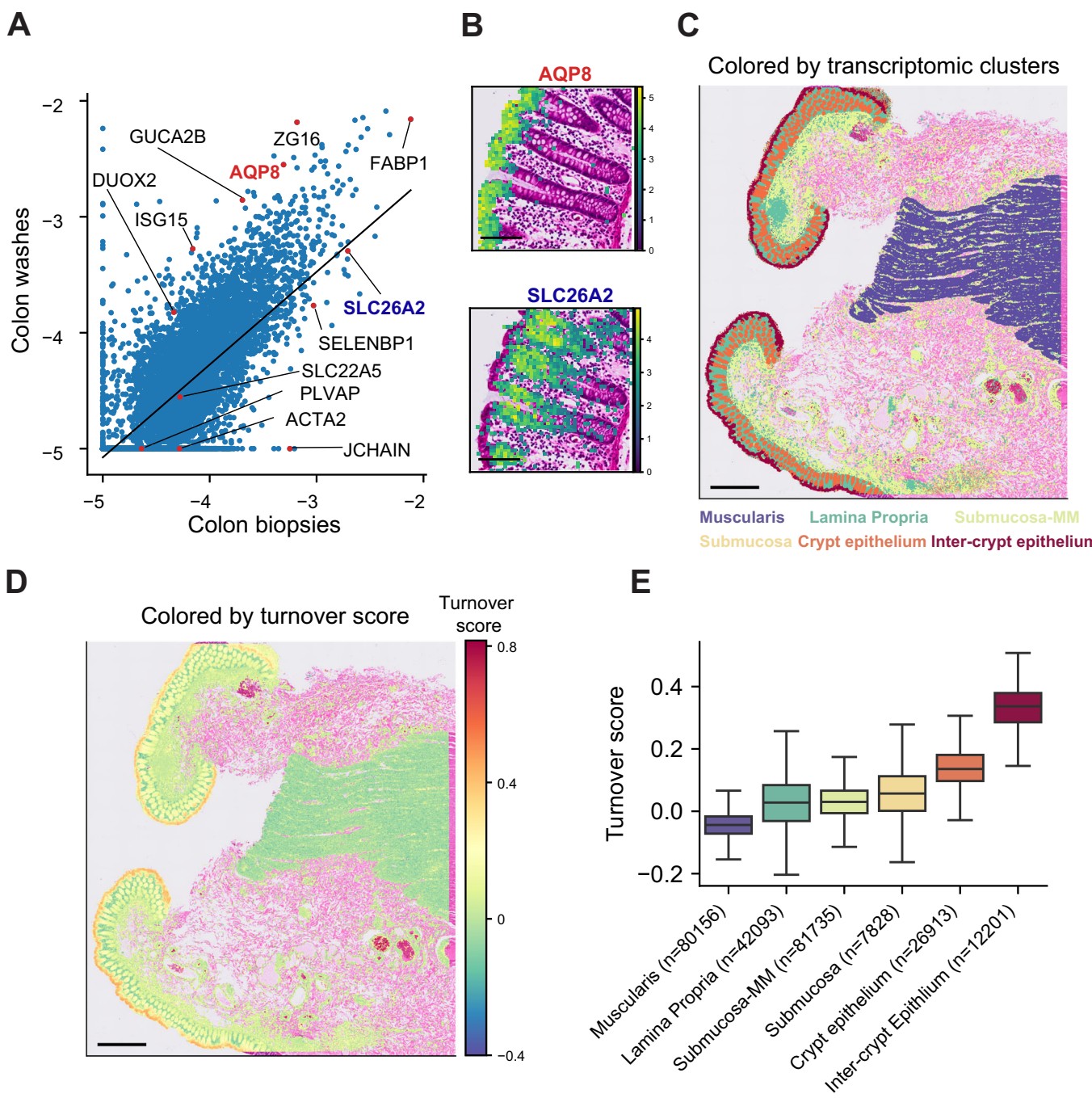

**Figure 4. Turnover maps of the human colon.**

(A) Scatter plot of gene expression in colonic fecal washes and colonic biopsies. Each dot is a gene; representative genes expressed more highly in washes or tissue are shown in red. Data taken from Dan et al (Dan et al, 2023). (B) Spatial expression patterns of *AQP8* (top) and *SLC26A2* (bottom) in the inter-crypt and intra-crypt epithelium, respectively. (C) VisiumHD spatial transcriptomics of the human colon colored by expression clusters (6-nearest neighbors), annotated according to the respective anatomical compartment. (D) Tissue colored by inferred turnover score. Microscopy and gene expression data in (B–E) from Oliveira et al (Oliveira et al, 2025), pixels with more than 20 UMIs are included in the analysis, scale bars in (B) are 100 μm, in (C, D) are 800 μm. (E). Box plot of turnover scores of the 6 anatomical compartments in (C). Box plots show the median as the center line, boxes span the 25th–75th percentiles, whiskers extend up to 1.5 IQR, and outliers are not shown. Turnover scores computed using Eq. (1). VisiumHD binned data and microscopy images were obtained from 10x genomics (https://www.10xgenomics.com/datasets/visium-hd-cytassist-gene-expression-libraries-of-human-crc). Image courtesy of 10x Genomics, Inc. MM-Muscularis Mucosa.

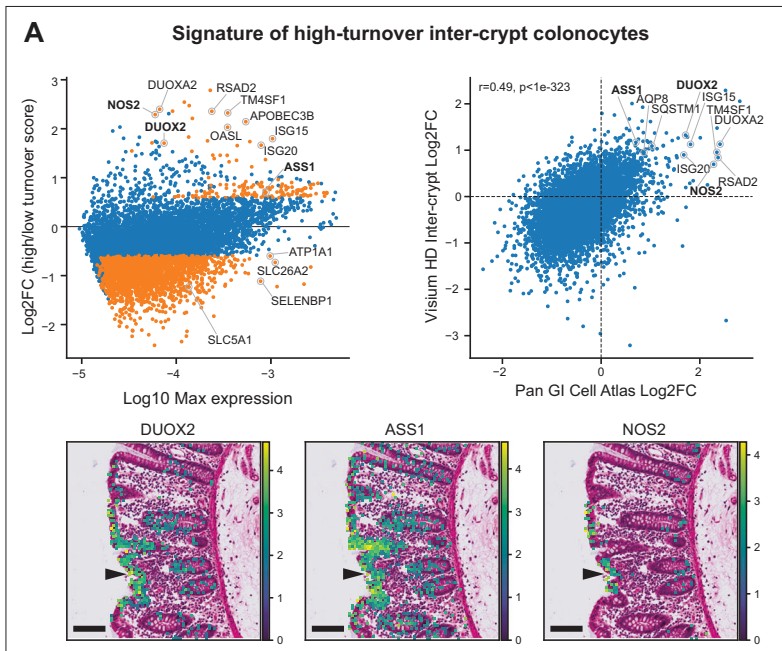

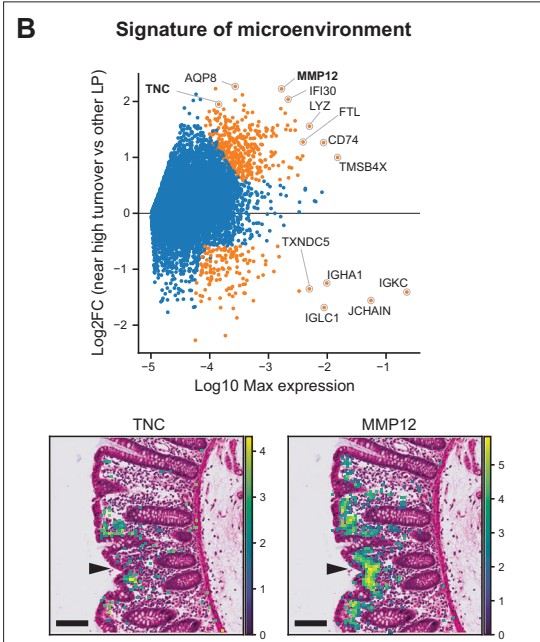

**Figure 5. Expression signatures and microenvironment of colonocytes with high turnover score.**

(A) Expression signature of high turnover inter-crypt colonocytes. Top left—MA plot of expression differences between single colonocytes with the top 33% and bottom 33% turnover scores. Single-cell data based on inter-crypt colonocytes from the Pan-GI Cell Atlas (Oliver et al, 2024). Top right—Fold changes between high and low turnover score pixels in colon spatial transcriptomics are correlated with fold changes observed in scRNAseq data. Microscopy and gene expression data is from Oliveira et al ((Oliveira et al, 2025), "Methods", external datasets). Bottom—Examples of spatial expression patterns of markers of high turnover colonocytes—DUOX2 (bottom left), ASS1 (bottom center), and NOS2 (bottom right). (B) MA plot of the differential gene expression between lamina propria pixels surrounding high (top 2%) and low (remaining 98%) turnover inter-crypt colonocytes (distance up to 70 µm from the respective pixels). Pixels with more than 100 UMIs are included in the analysis. Examples of spatial expression patterns of TNC and MMP12 are shown at the bottom. Arrowhead points toward a region with a high turnover score. All scale bars are 100 µm. (A, B) Orange dots denote genes with Benjamini–Hochberg q < 0.05, minimal fold change of 1.5, and maximal expression above 1e-5. VisiumHD binned data and microscopy image were obtained from 10x genomics (https://www.10xgenomics.com/datasets/visium-hd-cytassist-gene-expression-libraries-of-human-crc). Image courtesy of 10x Genomics, Inc. Expression levels are log-normalized.

## Spatial signatures of high turnover epithelial cells

We next focused on the colonic inter-crypt epithelium and asked whether cells that exhibit higher turnover within this anatomical compartment might carry distinct gene expression signatures. To this end, we performed differential gene expression analysis between the inter-crypt colonocytes with the highest and lowest turnover scores ("Methods", external datasets). We used two independent datasets – the pan-GI cell atlas (Oliver et al, 2024) and the colon Visium HD (Oliveira et al, 2025). Expression signatures of high turnover colonocytes significantly correlated between the two datasets (Fig. 5A). We found that colonocytes with higher turnover scores exhibited elevated levels of interferon response genes (ISG15, ISG20, RSAD2, and DUOX2, Fig. 5A). High turnover colonocytes also exhibited elevated levels of ASS1 and NOS2 (Fig. 5A). ASS1 encodes the enzyme Arginosuccinate Synthetase 1, which produces precursors of arginine, a substrate used by the inducible Nitric Oxide Synthase gene (NOS2) to produce nitric oxide (NO), which plays a central role in immune defense, inflammation, and signaling (Coleman, 2001; Bogdan, 2001). Increased NO production in colonocytes could therefore be associated with higher turnover.

We next sought to identify distinct signals representative of the surrounding tissue microenvironment of colonocytes with high turnover score. To this end, we compared the Visium HD expression of adjacent pixels of high and low turnover colonocytes ("Methods", external datasets). Lamina propria cells adjacent to high turnover colonocytes exhibited elevated levels of Tenascin, encoded by the gene TNC (Fig. 5B), a marker of human villus tip telocytes (Harnik et al, 2024) that has been shown to stimulate macrophage recruitment. Additional elevated genes in the vicinity of high turnover colonocytes included the macrophage elastase MMP12 (Fig. 5B), a central player in inflammatory processes in the colon (Nighot et al, 2021; Dufour et al, 2018) and in the liver (Ben-Moshe et al, 2022). Our approach therefore exposes signatures of high turnover cells and of associated microenvironmental signals.

## Discussion

Single-cell atlases and spatial transcriptomics have proven instrumental in characterizing cellular states in mammalian organs, yet they do not inform on dynamic cellular properties such as turnover rates. Our study utilizes the molecular information of shed cells to enable estimating turnover patterns of distinct cell types. This facilitates identification of expression states and tissue microenvironments associated with higher turnover. While we demonstrated our approach on the organs of the digestive tract in health, it is highly suited for additional organs and for disease states. Shed cell

transcriptomes in urine (Vorperian et al, 2024; Abedini et al, 2021) can be profiled to characterize turnover patterns in the kidneys and urogenital tract. Cell-free RNA includes transcripts from dying cells in diverse internal organs such as the liver (Vorperian et al, 2022), and could be used to explore hepatocyte turnover rates in health and in diverse pathologies. Our approach is particularly suited for the analysis of pathologies in the digestive tract. Celiac disease and inflammatory bowel diseases are associated with massive death of epithelial cells (Moss et al, 1996; Patankar and Becker, 2020). Estimating turnover patterns could reveal enterocyte and colonocyte sub-populations that are preferentially attacked by the immune system in these diseases. Reconstructing turnover maps of tumors, such as colorectal tumors, could highlight sub-clones or molecular states that are more fit to the limiting tumor microenvironment. Such low-turnover cells may be potential targets of novel therapies.

Our method has several potential limitations. It relies on the mRNA half-life of shed cells, which can vary with physiological states such as stress, diet, or disease. Like other RNA-based techniques, it may be sensitive to technical factors, from sample collection to processing. Disease-related shedding—triggered by conditions like neoplasia, necrosis, or inflammation—can alter the quantity and composition of the shed cells or cell fragments. In such cases, tailored sampling of both tissue and washes may improve accuracy. Biological factors add complexity: differentiation may reduce representation of certain cell states, and RNA release not associated with cellular shedding may also occur. Directionality of shedding (apical vs. basolateral) introduces lineage-specific biases—for example, limiting the detection of submucosal lineages in luminal samples. Thus, the method is particularly informative for epithelial lineages. Changes in gene expression after shedding (Bahar Halpern et al, 2023) could be taken into consideration in future developments of our approach.

Methodologically, turnover scores are derived from linear regression on bulk RNA data, which integrates signals from mixed populations. This can limit the resolution of cell-type-specific dynamics. Moreover, microenvironmental signatures may reflect correlations rather than causality. Despite these challenges, our turnover inference approach adds a dynamic layer to single-cell atlases, offering insights to diverse human tissues in health and disease.

## Methods

### Reagents and tools table

| Reagent/resource | Reference or source | Identifier or catalog number |
| --- | --- | --- |
| **Oligonucleotides and other sequence-based reagents** | | |
| Reverse transcriptase Maxima H | Thermo | EP0753 |
| Barcoded RT primer | IDT | |
| dNTPs | Thermo | R0182 |
| TSO *E5V6NEXT | IDT | |
| PEG8000 | Sigma | 89510 |
| AMPure XP beads | Beckman Coulter | A63881 |
| Exo1 | Thermo | EN0582 |
| Terra polymerase, enzyme, and buffer | Takara | 639270 |

| Reagent/resource | Reference or source | Identifier or catalog number |
| --- | --- | --- |
| Nextera XT library prep | Illumina | FC131-1024 |
| SINGV6 primer | IDT | |
| EB8 | Sigma | T2694 |
| Dual-indexed barcode | IDT | |
| **Chemicals, enzymes, and other reagents** | | |
| TRI Reagent | Sigma | T9424 |
| Direct-zol micro prep kit | ZYMO | R2062 |
| High Sensitivity D1000 ScreenTape | Agilent | 5067-5584 |
| TapeStation buffer | Agilent | 5067-5585 |
| **Software** | | |
| MATLAB | | |
| Python | | |
| UTAP | | |

## Methods and protocols

### Upper GI fluid collection

Patients arriving for a medical procedure were recruited without blinding at the general surgery department of Sheba Medical Center (Helsinki #SMC-8665-21). For all the patients, written informed consent was obtained in accordance with the Declaration of Helsinki, and the experiments conformed to the principles set out in the WMA Declaration of Helsinki and the Department of Health and Human Services Belmont Report. Patients were expected to undergo abdominal surgeries, for which NGT was placed regardless of patient participation in the study, or patients with indwelling NGT. Clinical and demographic data were obtained from patients' electronic health records. Samples from patients with evidence of upper GI system malignancy were not included in the current study. Upper GI tract fluids were collected directly from the NGTs upon the appearance of fluids with green hue, evidence for biliary secretion that should enrich for duodenal shed cells. All samples were processed within 20 min of collection. Samples were delivered to the lab in 15-mL tubes on ice. At the lab, samples were centrifuged at $580 \times g$ for 10 min at 4 °C, the supernatant was removed, and the pellet was frozen immediately on dry ice and kept in $-80$ °C until further processing.

### RNA extraction

Tri-reagent was added at a ratio of 1:3, and samples were thawed on ice and thoroughly mixed every few minutes. Samples with a significant presence of solid content underwent additional centrifugation at this stage at $500 \times g$ for 5 min at 4 °C. Ethanol was added at a ratio of 1:1 to the supernatant, and extraction was continued according to the manufacturer's instructions of Direct-zol micro prep kit (ZYMO research, R2062).

### Bulk RNA sequencing of the samples

The mcSCRBseq protocol (Bagnoli et al, 2018) was used with minor modifications. Reverse transcription was applied on up to 500 ng of RNA, with a final volume of 20 μL (1×Maxima hour Buffer, 1 mM

dNTPs, 2 µM TSO* E5V6NEXT, 7.5% PEG8000, 20U Maxima H enzyme, 2 µL barcoded RT primer). Subsequent steps were applied as mentioned in the protocol. Library preparation was performed using Nextera XT kit (Illumina) on 1–2.5 ng amplified cDNA. Library final concentration was 1.8–2 nM, and sequencing was done using the Novaseq 6000/X (Illumina) sequencing machine aiming at 40 M reads per sample with the following settings: Read1-16bp, Index1-10bp, Index2-10bp, Read2-66bp, few samples were sequenced with Index1-8bp, Index2-8bp.

The following primers were used:

- Barcoded RT primer: 5′-/5Biosg/ACACTCTTTCCCTACAC-GACGCTCTTCCGATCT[6-bp cell barcode][10-bp UMI] T$_{30}$VN -3′
- TSO *E5V6NEXT: ACACTCTTTCCCTACACGACGCrGrGrG
- SINGV6: /5Biosg/ACACTCTTTCCCTACACGACGC
- Dual-indexed barcode:
  - CAAGCAGAAGACGGCATACGAGAT[i7] GTCTCGTGGGCTCGG
  - AATGATACGGCGACCACCGAGATCTACAC[i5] TCGTCGGCAGCGTC.

### Bioinformatics and computational analysis of bulk RNA sequencing data

Illumina output sequencing raw files were converted to FASTQ files using the UTAP pipeline (Kohen et al 2019). To obtain the UMI counts table, UTAP for SCRB-seq was used with CUTA-DAPT, and FASTQ files were aligned to the human reference genome (GRCh38). Data filtration and statistical analysis were performed with MATLAB R2024a and Python 3.9 using Scanpy 1.9.3 packages. Only protein-coding genes were retained, and mitochondrial and ribosomal genes were removed. Biomart for reference genome GRch38 V91 was used for gene classification as protein-coding. A total of 31 NGT samples with more than 10,000 UMIs were included in the analysis. Gene expression for each sample was normalized by the sum of the UMIs of the remaining genes.

### Assessment of the reproducibility of NGT fluids

For the assessment of reproducibility (Fig. EV2A), we sampled three patients twice, at time intervals of 10–15 min. We created a clustergram based on the Z-transformed sum-normalized expression. Only genes with median expression above 1e-4 were included in the analysis. Two of the three patients suffer from malignancy (120 from gastric cancer and 125 from gallbladder cancer), thus they were excluded from our primary analysis.

### Calculation of turnover score

For normalization, UMI counts of each wash and biopsy were divided by the sum of UMIs of that sample. The log10 medians of wash normalized expression plus a pseudo number of 1e-5 were linearly regressed against the log10 median of biopsy normalized expression, with a constant term added to allow estimation of an intercept, yielding residuals for each gene. Positive residuals were genes that were higher in washes, whereas negative residuals were genes that were higher in biopsies. Turnover score of a cell/pixel was defined as the sum of the residuals weighted by the UMI summed-normalized gene expression (Eq. (1)).

### Calculation of tip score

Villus tip and bottom markers were taken from Moor et al (Moor et al, 2018), each single-cell was scored using the sum of the tip markers divided by the sum of the tip and bottom markers.

$$Tip\ score = \frac{\sum Tip\ markers}{\sum Tip\ markers + \sum Bottom\ markers}$$

### External datasets

This study used data from Moor et al (Moor et al, 2018), Bahar Halpern et al, (Bahar Halpern et al, 2023), GTEX (Lonsdale et al, 2013), Abadie et al, (Abadie et al, 2020), Busslinger et al, (Busslinger et al, 2021), Pan-GI cell atlas (Oliver et al, 2024), Oliveria et al, (Oliveira et al, 2025) and Dan et al, (Dan et al, 2023). Tabula Sapiens (The Tabula Sapiens Consortium, 2022), Elmentaite et al (Elmentaite et al, 2021), and Chalassani et al (Chalasani et al, 2021).

Single shed-cells RNA sequencing data, and mouse jejunal bulk biopsies and washes were taken from Bahar Halpern et al (Bahar Halpern et al, 2023), residuals were calculated as detailed above. Apoptotic genes taken from HALLMARK_APOPTOSIS gene set v2024.1. For Fig. EV1C,D, the turnover score was calculated as detailed above, with and without apoptotic genes, respectively.

Single-cell RNA sequencing of mouse small intestine epithelium was taken from Moor et al (Moor et al, 2018). Human esophagus and stomach bulk gene expression was extracted from GTEX (Lonsdale et al, 2013). Human duodenum bulk gene expression was extracted from Abadie et al (Abadie et al, 2020), including control patients above 30 years of age. Single-cell data of human esophagus, stomach, and duodenum were obtained from Busslinger et al (Busslinger et al, 2021). Cells included had UMI counts ranging from 1500 to 10,000, and mitochondrial fraction below 50%. We used the original cellular annotations, after filtration we excluded cell types with less than 15 cells except goblet and Paneth cells.

Estimation of residuals for the human colon datasets was based on Dan et al (Dan et al, 2023). Samples included were biopsies and fecal washes from healthy controls from the right and left colon, with a minimum of 10,000 UMIs per sample. Mitochondrial and ribosomal genes were excluded. Single-cell analysis of the human colon was based on the pan-GI atlas for epithelial cells of the large intestine of control patients (Oliver et al, 2024). Inter-crypt epithelial cells were defined as cells from the large intestine of control subjects that were annotated as epithelial cells (level 1 annotation) with expression of the inter-crypt markers AQP8 and GUCA2B above 1e-3 and expression of the goblet cell marker ZG16 below 1e-3. Inter-crypt colonocytes with turnover scores in the top 33% and in the bottom 33% were compared. Spatial transcriptomics of the human colon consisted of non-tumor colon tissue from patient P3 from Oliveira et al (FFPE section of normal adjacent colon tissue from an 83-year-old male with a transverse colon tumor (VisiumHD, Space Ranger v3.0, 10x Genomics)). We identified inter-crypt epithelium spots by intersecting the annotation of the whole mucosa, K-Nearest Neighbors clustering of the inter-crypt epithelium (n = 6, taken from Loupe browser, 10X) that was refined according to AQP8 expression of 2e-3. Lamina propria pixels surrounding high (top 2%) and low (remaining 98%) turnover inter-crypt colonocytes with a distance up to 70 µm from respective pixels with more than 100 UMIs were included in the analysis of Fig. 5B.

Identification of large intestine and blood-specific genes was based on the identification of organ-specific genes from the signature matrix of Tabula Sapiens (The Tabula Sapiens Consortium, 2022), with minimal sum-normalized expression of 1e-5, and minimal fold-change of 1.5 with pseudo number of 1e-5. cfRNA RNA sequencing data was taken from healthy individuals from Chalasani et al (Chalasani et al, 2021). Identification of colonic-specific cell types marker genes was based on Elmentaite et al (Elmentaite et al, 2021) with minimum cell types count of 300 cells except enteroendocrine cells, markers were identified with minimal expression of 1e-5, and minimal fold-change of 1.5 with pseudo number of 1e-5. For each cell type, mean and SEM of the log10 sum-normalized expression for the non-zero markers (either at the colonic wash or at the serum cfRNA) were calculated.

## Data availability

The dataset and computer code produced in this study are available in the following databases: Source code is available at Zenodo (https://zenodo.org/records/15706697) and Github (https://github.com/barkait/TurnoverScore). Bulk RNA-Sequencing data: Count matrices of the NGT samples and corresponding metadata are available as Datasets (Datasets EV1 and EV2, respectively) and in the Gene Expression Omnibus (GEO) under accession number GSE301268. Due to ethical restrictions and patient confidentiality, the raw sequencing data are not publicly available.

The source data of this paper are collected in the following database record: biostudies:S-SCDT-10_1038-S44320-025-00154-w.

## Peer review information

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

## Acknowledgements

SI is supported by the Moross Integrated Cancer Center, the Helen and Martin Kimmel Award for Innovative Investigation, the Yad Abraham Research Center for Cancer Diagnostics and Therapy, the Israel Science Foundation grants no. 908/21 and no. 3663/21, the European Research Council (ERC) under the European Union's Horizon 2020 research and innovation programme, grant no. 768956, a Weizmann-Sheba joint research grant and a research grant from the Ministry of Innovation, Science and Technology, Israel. Illustrations in Figs. 1, 2C, 3A, I–J were created using BioRender.

## Author contributions

**Tal Barkai**: Conceptualization; Data curation; Formal analysis; Methodology; Writing—original draft; Writing—review and editing. **Oran Yakubovsky**: Data curation; Methodology. **Yael Korem Kohanim**: Formal analysis. **Keren Bahar Halpern**: Data curation; Methodology. **Sapir Shir**: Formal analysis. **Noa Oren**: Data curation. **Michal Fine**: Data curation. **Paz Kelmer**: Data curation. **Amit Talmon**: Data curation. **Alon Israeli**: Data curation. **Niv Pencovich**: Data curation. **Ron Pery**: Data curation. **Ido Nachmany**: Supervision. **Shalev Itzkovitz**: Conceptualization; Formal analysis; Supervision; Funding acquisition; Writing—original draft; Writing—review and editing.

Source data underlying figure panels in this paper may have individual authorship assigned. Where available, figure panel/source data authorship is listed in the following database record: biostudies:S-SCDT-10_1038-S44320-025-00154-w.

## Disclosure and competing interests statement

The authors declare no competing interests.

# Expanded View Figures

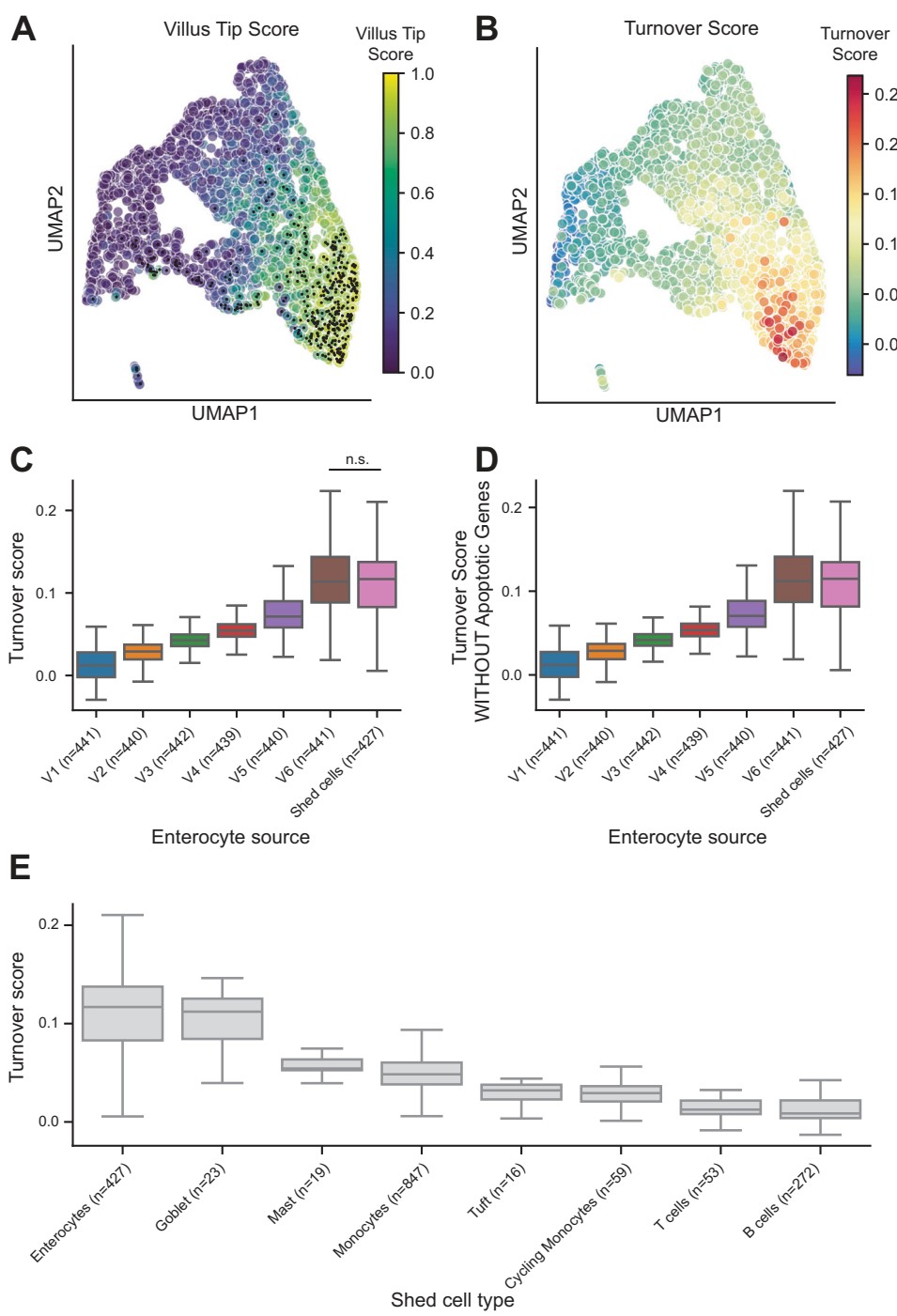

**Figure EV1. Turnover scores of single cells from mouse small intestine.**

(A) UMAP displaying tissue enterocytes and shed cells, with shed cells marked with black dots. (B) UMAP colored by Turnover score. (C) Turnover score of each zone, apoptotic genes included, there is no significant difference between V6 and shed-cells ($P = 0.1$, Wilcoxon rank-sum test). (D) Turnover score of each zone and shed cells without apoptotic genes. (E) Turnover score of all shed cells types from Bahar Halpern dataset (Bahar Halpern et al, 2023). Box plots show the median as the center line, boxes span the 25th–75th percentiles, whiskers extend up to 1.5 IQR, outliers are not shown.

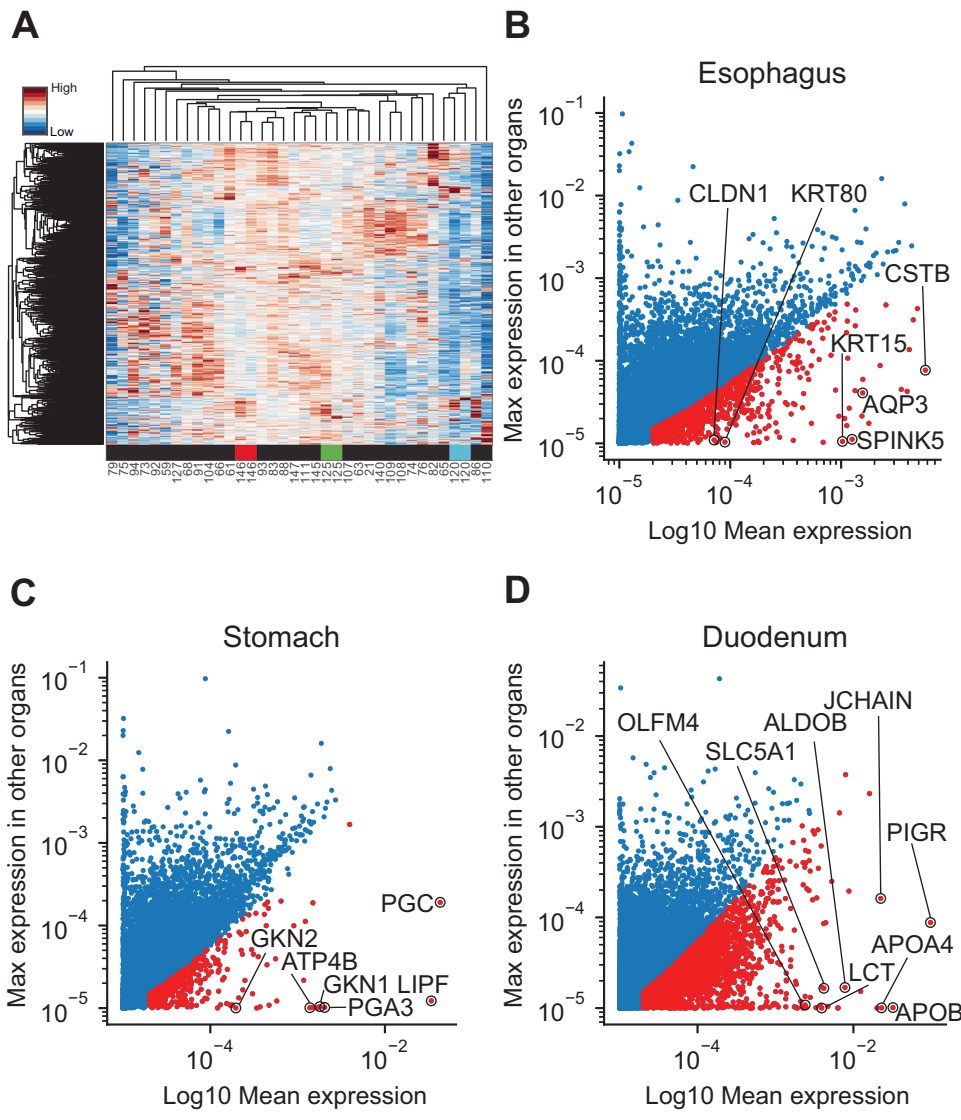

**Figure EV2. Reproducibility and marker stratification of NGT fluids.**

(A) Clustergram of NGT samples shows reproducibility across patients. Every column represents a sample and every row represents a gene. Colors denote biological repeats obtained at 10–15 min intervals. (B–D) Marker gene selection of the analyzed tissues—Esophagus (B), Stomach (C), and Duodenum (D). Red dots denote genes with maximal expression above 1e-5 and fold-change above 2 compared to the remaining two organs. Representative genes are highlighted. A pseudo-number of 1e-5 was added to the data.

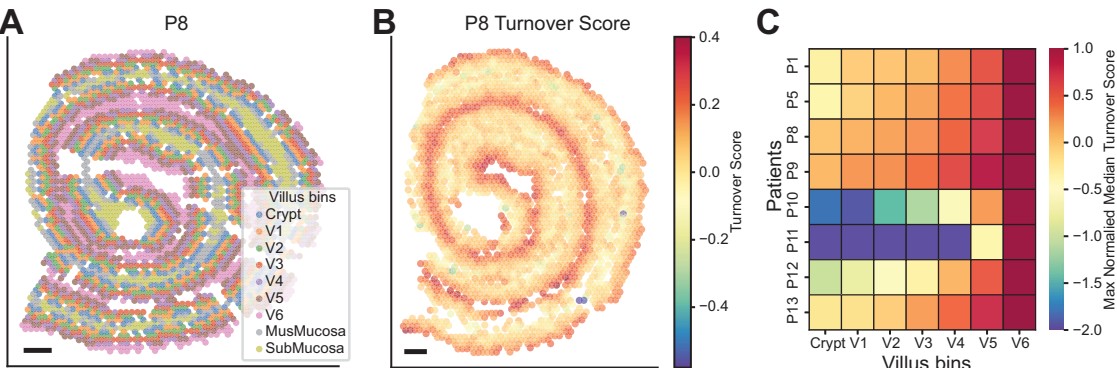

**Figure EV3. Spatial turnover maps of the human small intestine.**

(A) Annotated Visium dataset of the human small intestine from Harnik et al (Harnik et al, 2024). (B) Visium data colored by turnover score. Scale bars in (A, B) are 500 μm. (C) Average turnover score of crypt–villus zones for the 8 patients analyzed in Harnik et al (Harnik et al, 2024). Values lower than −2 were set to −2.

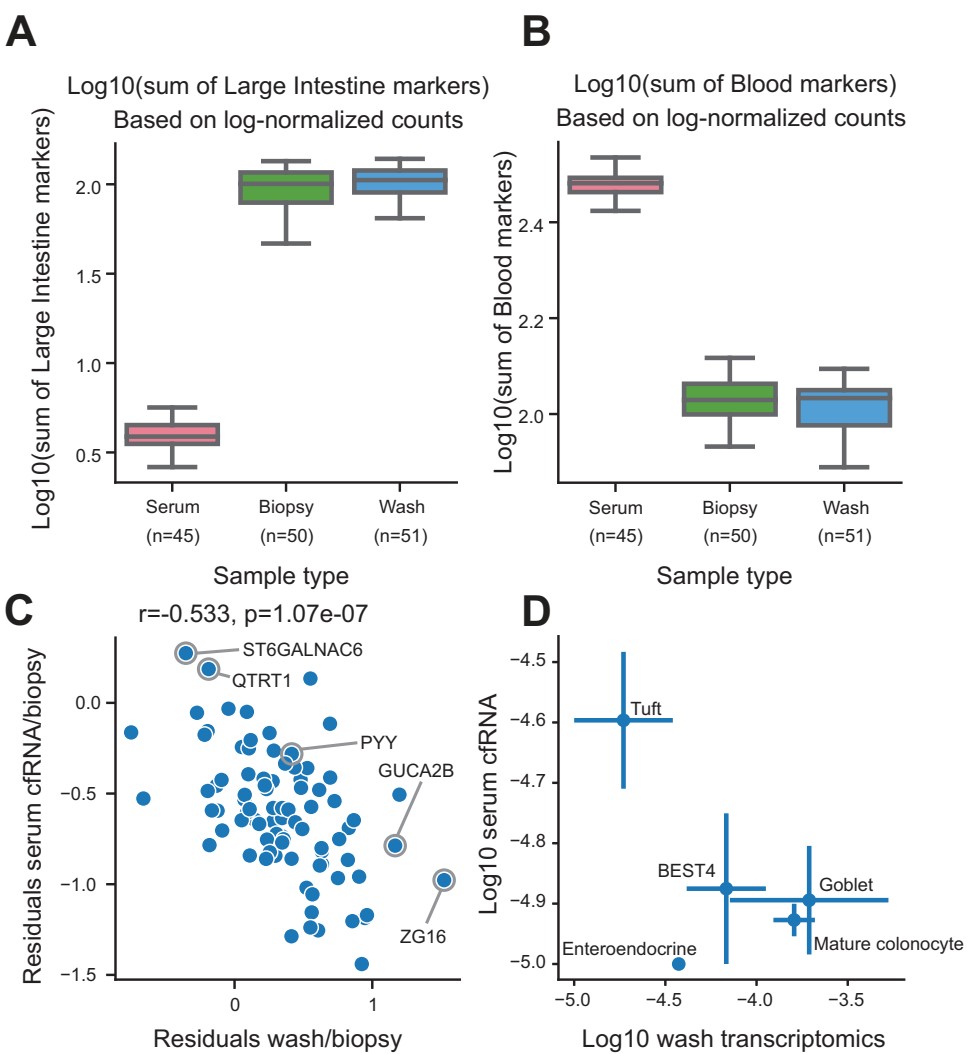

**Figure EV4.  Luminal shedding is different from basolateral shedding.**

(A) Sum of Large Intestine markers (log-normalized counts) of serum, biopsy and wash. (B) Sum of Blood markers of serum, biopsy and wash, serves as a positive control for (A). (C) Scatter plot of residuals of expressed genes in both wash and biopsy, representative genes are shown. (D) Mean large intestine cell type markers expression in serum cfRNA and wash transcriptomics. The number of genes were 22 in mature enterocytes, 6 in Goblet cells, 4 in Tuft cells, 2 in BEST4 cells and 1 in Enteroendocrine cells. Box plots show the median as the center line, boxes span the 25th–75th percentiles, whiskers extend in (A–C) extend up to 1.5 IQR, outliers are not shown. Whiskers in (D) represent SEM.

