## [Peer Review File · Molecular Systems Biology]

Transcriptomic profiling of shed cells enables spatial mapping of cellular turnover in human organs

Tal Barkai, Oran Yakubovsky, Yael Korem Kohanim, Keren Bahar Halpern, Sapir Shir, Noa Oren, Michal Fine, Paz Kelmer, Amit Talmon, Alon Israeli, Niv Pencovich, Ron Pery, Ido Nachmany, and Shalev Itzkovitz

Corresponding author(s): Shalev Itzkovitz (shalev.itzkovitz@weizmann.ac.il)

Review Timeline:

Submission Date:	17th Mar 25
Editorial Decision:	5th May 25
Revision Received:	24th Jul 25
Editorial Decision:	18th Aug 25
Revision Received:	20th Aug 25
Accepted:	18th Sep 25

Editor: Poonam Bheda

Transaction Report:

5th May 2025

Manuscript Number: MSB-2025-12964

Title: Transcriptomic profiling of shed cells enables spatial mapping of cellular turnover in human organs

Dear Dr. Itzkovitz,

Thank you again for submitting your work to Molecular Systems Biology. We have now heard back from the three reviewers who agreed to evaluate your study. As you will see below, the reviewers appreciate that the proposed approach addresses a timely topic. However, they raise a series of concerns, which we would ask you to address in a major revision.

Without repeating all the comments listed below, some of the more fundamental issues raised are the following:

- Additional factors that could affect turnover potential should be addressed (ideally via discussion and experimental analysis), and any limitations of the study should be discussed
- In line with comments from Reviewers 1 and 3, the code must be deposited in a publicly accessible repository such as Github. It is not sufficient to have the code available upon request.

All other issues raised would need to be satisfactorily addressed. Please let me know in case you would like to discuss in further detail any of the comments, I would be happy to schedule a call.

We require:

1) A .docx formatted version of the manuscript text (including legends for main figures, EV figures and tables). Please make sure that the changes are highlighted to be clearly visible. Alternatively you may choose to submit your manuscript as a LaTeX file.

4) A .docx formatted letter INCLUDING the reviewers' reports and your detailed point-by-point responses to their comments. As part of the EMBO Press transparent editorial process, the point-by-point response is part of the Peer Review File (PRF), which will be published alongside your paper.

5) A complete author checklist, which you can download from our author guidelines (<https://www.embopress.org/page/journal/17574684/authorguide#submissionofrevisions>). Please insert information in the checklist that is also reflected in the manuscript. The completed author checklist will also be part of the PRF.

6) Please note that all corresponding authors are required to supply an ORCID ID for their name upon submission of a revised manuscript.

7) It is mandatory to include a 'Data Availability' section after the Materials and Methods. Before submitting your revision, primary datasets produced in this study need to be deposited in an appropriate public database, and the accession numbers and database listed under 'Data Availability'. Please remember to provide a reviewer password if the datasets are not yet public (see <https://www.embopress.org/page/journal/17574684/authorguide#dataavailability>).

In case you have no data that requires deposition in a public database, please state so in this section as follows: "This study includes no data deposited in external repositories". Note that the Data Availability Section is restricted to new primary data that are part of this study.

8) All Materials and Methods need to be described in the main text using our 'Structured Methods' format, which is required for all research articles. According to this format, the Methods section includes a Reagents and Tools Table (listing key reagents, experimental models, software and relevant equipment and including their sources and relevant identifiers) followed by a Methods and Protocols section describing the methods using a step-by-step protocol format. The aim is to facilitate adoption of the methodologies across labs. Please upload the Reagents and Tools table as a separate document when submitting your revised manuscript. More information on how to adhere to this format as well as a downloadable template (.docx) for the

Reagents and Tools Table can be found in our author guidelines:
<https://www.embopress.org/page/journal/17444292/authorguide#structuredmethods>

An example of a Method paper with Structured Methods can be found here:
<https://www.embopress.org/doi/10.15252/msb.20178071>.

9) For data quantification: please specify the name of the statistical test used to generate error bars and p-values, the number (n) of independent experiments (specify technical or biological replicates) underlying each data point and the test used to calculate p-values in each figure legend. The figure legends should contain a basic description of n, p-values and the test applied. Graphs must include a description of the bars and the error bars (s.d., s.e.m.). Please provide exact p-values (in either the figure or figure legend).

10) Our journal encourages inclusion of *data citations in the reference list* to directly cite datasets that were re-used and obtained from public databases. Data citations in the article text are distinct from normal bibliographical citations and should directly link to the database records from which the data can be accessed. In the main text, data citations are formatted as follows: "Data ref: Smith et al, 2001" or "Data ref: NCBI Sequence Read Archive PRJNA342805, 2017". In the Reference list, data citations must be labeled with "[DATASET]". A data reference must provide the database name, accession number/identifiers and a resolvable link to the landing page from which the data can be accessed at the end of the reference. Further instructions are available at .

11) We replaced Supplementary Information with Expanded View (EV) Figures and Tables that are collapsible/expandable online. EV Figures should be cited as 'Figure EV1, Figure EV2' etc... in the text and their respective legends should be included in the main text after the legends of regular figures.

- Additional Tables/Datasets should be labeled and referred to as Table EV1, Dataset EV1, etc. Legends should be provided in a separate tab in case of .xls files. Alternatively, the legend can be supplied as a separate text file (README) and zipped together with the Table/Dataset file.

<https://www.embopress.org/page/journal/17574684/authorguide#expandedview>

12) Author contributions: CRediT has replaced the traditional author contributions section because it offers a systematic machine-readable author contributions format that allows for more effective research assessment. Please remove the Authors Contributions from the manuscript and use the free text boxes beneath each contributing author's name in our system to add specific details on the author's contribution. More information is available in our guide to authors.

13) Disclosure statement and competing interests: We updated our journal's competing interests policy in January 2022 and request authors to consider both actual and perceived competing interests. Please review the policy
<https://www.embopress.org/competing-interests> and update your competing interests if necessary.

14) Every published paper now includes a 'Synopsis' to further enhance discoverability. Synopses are displayed on the journal webpage and are freely accessible to all readers. They include a short stand first (maximum of 300 characters, including space) as well as 2-5 one-sentences bullet points that summarizes the paper. Please write the bullet points to summarize the key NEW findings. They should be designed to be complementary to the abstract - i.e. not repeat the same text. We encourage inclusion of key acronyms and quantitative information (maximum of 30 words / bullet point). Please use the passive voice. Please attach these in a separate file or send them by email, we will incorporate them accordingly.

Please note that these would be the final versions and changes during proofing are usually not allowed.

15) As part of the EMBO Publications transparent editorial process initiative (see our policy here:
https://www.embopress.org/transparent-process#Review_Process), Molecular Systems Biology will publish online a Peer Review File (PRF) to accompany accepted manuscripts.

In the event of acceptance, this file will be published in conjunction with your paper and will include the anonymous referee reports, your point-by-point response and all pertinent correspondence relating to the manuscript. Let us know whether you agree with the publication of the PRF and as here, if you want to remove or not any figures from it prior to publication. Please note that the Author checklist will be published at the end of the PRF.

Molecular Systems Biology has a "scooping protection" policy, whereby similar findings that are published by others during review or revision are not a criterion for rejection. Should you decide to submit a revised version, I do ask that you get in touch after three months if you have not completed it, to update us on the status.

Yours sincerely,

Poonam Bheda, PhD
Scientific Editor
Molecular Systems Biology

Reviewer #1:

Summary

This manuscript by Barkai et al. presents an innovative approach to infer cellular turnover rates in human gastrointestinal (GI) tissues by integrating transcriptomic data from shed cells with spatially resolved single-cell atlases. The authors develop a turnover scoring method based on the over- or under-representation of gene expression in shed cells compared to tissue samples. They apply this approach to different compartments of the upper and lower human GI tract, as well as the mouse intestine, and validate their turnover maps against known anatomical and biological patterns. Of particular note is the identification of short-lived, interferon-stimulated colonocytes in the human colon, which are associated with a distinct pro-inflammatory microenvironment, and might be therefore of clinical relevance.

General remarks

The manuscript by Barkai et al. convincingly supports its key conclusions through robust integration of shed cell transcriptomics, spatial single cell atlases and thorough validation across species and GI compartments. The study introduces a conceptually elegant and straight-forward method to infer cellular turnover rates - a dynamic tissue property that has been difficult to quantify in humans without invasive or longitudinal approaches. By using naturally shed cells and combining them with spatial data, the authors offer a scalable, non-invasive strategy that overcomes ethical and technical barriers, particularly in healthy individuals, though still requiring swabs.

This work represents a conceptual and methodological advance in the fields of tissue dynamics and single-cell genomics. While single-cell atlases have elucidated cellular identities and states, they often lack dynamic information such as proliferation or death rates. This study elegantly fills this gap, providing a new dimension for interpreting single-cell data in a dynamic physiological context. Compared to previous knowledge, which relied primarily on model organisms, genetic labelling or indirect markers of turnover, this approach allows direct inference of turnover patterns in human tissues at high resolution, validated against established anatomical and biological benchmarks.

This advance will be of great interest to a wide audience, including researchers in systems biology, gastroenterology, immunology, computational biology and regenerative medicine. In addition, the potential applicability of the method to disease contexts such as inflammatory bowel disease, cancer and epithelial barrier disorders further enhances its relevance to clinical and translational researchers seeking dynamic biomarkers or therapeutic targets.

Major points

1. Dynamic transcriptional states of shed cells:

A key consideration is the potential transcriptional changes that cells may undergo during or after shedding. Stress responses, apoptosis or post-shed degradation may affect gene expression profiles, potentially introducing bias into turnover score estimates. While this is not a fundamental limitation of the approach, the authors should address how such dynamic processes may affect the interpretation of their turnover scores and whether specific gene signatures (e.g. stress or apoptosis markers) were evaluated, or regressed out, in this context. Furthermore, (gene-)specific transcriptional turnover rates are not considered but could further refine the mathematical model. The authors could try to incorporate a transcriptome-wide weighting system to correct for different mRNA half-lives, and thereby also learn something about the shedding dynamics (during sampling).

2. Sample variability and reproducibility:

The turnover scoring approach relies on samples obtained during surgical procedures or colonoscopies. Variability in patient conditions (e.g. surgical stress, diet) or technical handling may affect RNA profiles. The authors should clarify whether technical replicates, patient-matched samples or independent cohorts have been evaluated to assess the robustness of turnover scores (by revisiting their published data).

3. Previous attempts to model cellular dynamics from scRNA-seq data:

The current study focuses on using static transcriptomic snapshots to infer dynamic properties such as turnover with a very straightforward approach. Previous studies have similarly aimed to reconstruct cellular dynamics from scRNA-seq data using computational models, including those based on trajectory inference, barcoding or deterministic modelling frameworks. It would strengthen the manuscript to acknowledge these previous efforts and clearly position the current approach in relation to them.

4. Generalisability to disease contexts:

The authors mention potential applications in diseases such as IBD and cancer. However, disease-associated shedding events (e.g. necrosis, immune-mediated epithelial injury) may be fundamentally different from physiological turnover. This could affect the interpretation of turnover scores. The authors should briefly discuss limitations and necessary considerations when applying the method to diseased tissues, particularly in inflammatory or neoplastic contexts.

5. Availability of code and reproducibility:

While the authors mention that the code is available on request, and the mathematical formalism is rather straight-forward to implement, providing public access (e.g. via a publicly available repository such as GitHub) to a well-documented pipeline would greatly improve data and method sharing according to the FAIR principles.

Minor points

- The Introduction would benefit from a brief reference to previous studies describing gene expression differences in apoptotic or shed epithelial cells.
- The authors should provide more information on sample sizes and different scales of observed effects between samples/tissues (fold-changes, absolutes of turnover score)

Reviewer #2:

This article 'Transcriptomic profiling of shed cells enables spatial mapping of cellular turnover in human organs' by Barkai et al. describes how cell turnover rates can be determined by comparing RNA expression profile of single cells in the gastrointestinal tract using bulk sequencing data generated from shed cells. The authors built on previous work from their lab where they elegantly demonstrated that shed cells can be used for bulk and single cell RNA transcriptomic read-outs in order to determine epithelial cell states. Extending this approach to determine cellular turnover rates would allow identification of aberrant turnover rates in specific gastrointestinal lineages in pathological contexts such as IBD and cancer, and could hold diagnostic value. This reviewer has the following questions / concerns that would need to be addressed:

The authors describe cellular turnover (here defined as 'the time between a cell's birth and death') using the difference between RNA levels in shed cells versus cells in tissue context. However, one might argue that there are additional factors influencing 'turnover potential' measured in this way, These include the loss of any cellular state due to differentiation into another state, differential release of RNA from different cell states and preferential apical versus basolateral shedding of particular lineages (e.g. Paneth cells may bias for basolateral clearance, all submucosal lineages in this study are not expected to lose most of their RNA to the intestinal lumen).

The authors should discuss these points. This can be partly experimentally addressed by performing single cell RNA/nuclei sequencing on shed cells directly allowing the comparison of cell type ratios.

Although this may be beyond the scope of the study, an extensive shed single cell dataset may allow for more granular statements on cell turnover rates (e.g. for rare lineages like Tuft cells, M cells, enteroendocrine cell subtypes).

Minor points

Some genes display somewhat unexpected residual patterns. *Lgr5* (stem cell gene) and *Defa22* (Paneth cell gene) are barely depleted in the shed fraction (Figure 2A), while *LCT* (enterocyte gene) surprisingly is among the depleted ones for the human duodenum (Figure 3D): Could the authors comment on this, could this deviation from the expected patterns e.g. correlated to expression levels?

In line with the previous point: Stem cells are predicted to have higher turnover scores than intestinal goblet cells (Figure 3G), which is not expected.

In Figure 5, the authors identify a population of colonocytes that harbors an increased turnover rate. Could these gene sets also be induced after shedding already happened (e.g. through exposed TLR receptors)? A single cell comparison between shed cells and colonocytes could reveal this, if post-shedding signatures are induced it is expected that unique signatures at a single cell level would appear

Reviewer #3:

In their manuscript, Barkai et al. aim to estimate cellular turnover rates at single-cell resolution by integrating bulk RNA-seq from tissue and shed cells, along with both single-cell and spatial transcriptomic data. Turnover rates in the human esophagus, stomach and small intestine are inferred based on RNA sequencing on shed cells from the upper gastrointestinal tract, in addition to inference of turnover potential in the human large intestine based on colonic fecal washes. The study provides a map of cellular turnover rates across tissues, cell types, and tissue microenvironments that potentially promote cell death. One intriguing finding is a subset of short-lived, interferon-stimulated colonocytes within a pro-inflammatory microenvironment.

This work advances our ability to better understand and characterize turnover rates in cellular populations, in a comparative manner, which is an important and timely challenge, with implications for healthy and pathological tissue states. The data generated in this study could also be used as a valuable resource for future studies. A few comments are elaborated below.

Comments:

- In the first section of the results, the process by which turnover scores are computed should be explained more clearly (consider adding relevant equations?). Since this is the basis for the entire study, it is crucial to clarify this point.
- Have you tried assigning turnover potential scores per cell directly at the cellular level and not via the gene-based analysis (which has its limitations, for example due to the assumption of linear combination of residuals).
- The Discussion section should also include discussion of limitations of the study, including the linearity assumption in the calculation of turnover scores, microenvironmental signatures showing correlation and not causal effects (without additional perturbations for example), the fact that the whole calculation of turnover scores is done by regressing over bulk measurements which may include convolution of signals coming from different cells, etc.
- It's better to have the code publicly available (e.g. on GitHub) and not only make it available upon request.
- There were several references to Methods sections from the main text which do not seem to correspond to any of the existing methods sections (e.g. in the Results subsection 'Spatial signatures of high turnover epithelial cells').
- Are there single-cell measurements of shed cells? If so, would it have been possible to map these directly to the spatial data or single-cell data to identify potential for shedding?
- Typo: 'Our approach therefore expose

Reviewer #1:

Summary

This manuscript by Barkai et al. presents an innovative approach to infer cellular turnover rates in human gastrointestinal (GI) tissues by integrating transcriptomic data from shed cells with spatially resolved single-cell atlases. The authors develop a turnover scoring method based on the over- or under-representation of gene expression in shed cells compared to tissue samples. They apply this approach to different compartments of the upper and lower human GI tract, as well as the mouse intestine, and validate their turnover maps against known anatomical and biological patterns. Of particular note is the identification of short-lived, interferon-stimulated colonocytes in the human colon, which are associated with a distinct pro-inflammatory microenvironment, and might be therefore of clinical relevance.

General remarks

The manuscript by Barkai et al. convincingly supports its key conclusions through robust integration of shed cell transcriptomics, spatial single cell atlases and thorough validation across species and GI compartments. The study introduces a conceptually elegant and straight-forward method to infer cellular turnover rates - a dynamic tissue property that has been difficult to quantify in humans without invasive or longitudinal approaches. By using naturally shed cells and combining them with spatial data, the authors offer a scalable, non-invasive strategy that overcomes ethical and technical barriers, particularly in healthy individuals, though still requiring swabs.

This work represents a conceptual and methodological advance in the fields of tissue dynamics and single-cell genomics. While single-cell atlases have elucidated cellular identities and states, they often lack dynamic information such as proliferation or death rates. This study elegantly fills this gap, providing a new dimension for interpreting single-cell data in a dynamic physiological context. Compared to previous knowledge, which relied primarily on model organisms, genetic labelling or indirect markers of turnover, this approach allows direct inference of turnover patterns in human tissues at high resolution, validated against established anatomical and biological benchmarks.

This advance will be of great interest to a wide audience, including researchers in systems biology, gastroenterology, immunology, computational biology and regenerative medicine. In addition, the potential applicability of the method to disease contexts such as inflammatory bowel disease, cancer and epithelial barrier disorders further enhances its relevance to clinical and translational researchers seeking dynamic biomarkers or therapeutic targets.

Major points

1. Dynamic transcriptional states of shed cells:

A key consideration is the potential transcriptional changes that cells may undergo during or after shedding. Stress responses, apoptosis or post-shed degradation may affect gene expression profiles, potentially introducing bias into

turnover score estimates. While this is not a fundamental limitation of the approach, the authors should address how such dynamic processes may affect the interpretation of their turnover scores and whether specific gene signatures (e.g. stress or apoptosis markers) were evaluated, or regressed out, in this context. Furthermore, (gene-)specific transcriptional turnover rates are not considered but could further refine the mathematical model. The authors could try to incorporate a transcriptome-wide weighting system to correct for different mRNA half-lives, and thereby also learn something about the shedding dynamics (during sampling).

We thank the reviewer for these excellent suggestions. We have now applied our methodology with and without apoptosis genes. In the new EV 1C&D panels we demonstrate that the results remain qualitatively similar, with significant ($p < 1e-323$) spearman correlation of 0.99 between cells' turnover score with and without apoptosis genes.

We added the following text to page 4:

“To validate the zonal trend in enterocyte turnover we further examined the turnover scores of a recently generated combined single cell atlas of tissue and shed cells in the mouse intestine (Bahar Halpern, EV1A,B). This data demonstrated that single shed enterocytes were similar to villus tip enterocytes both in their general transcriptome, as well as in their turnover scores (EV1C). Since shed cells may exhibit death-related changes in gene expression upon shedding, we repeated our shedding score calculation while excluding apoptotic genes (Methods). We found that the zonal trends in turnover scores remained unchanged (EV 1C-D). We further compared turnover of other mouse jejunal lineages and found that enterocytes and goblet cells showed high turnover scores, whereas immune cells and tuft cells showed relatively lower turnover scores (EV 1E), consistent with their anatomical distance from the lumen.”

We have now also evaluated the impact of transcript half-lives on the residuals of shed cell vs. tissue gene expression (Figure R1). We find that there is no significant correlation between the mRNA half-life and the residual. In addition, the harsh luminal environment of the gut is predicted to lower the mRNA half-life even more, reducing the effect of the mRNA stability on the results.

Figure R1. mRNA half-life does not correlate with tissue-shed cell fraction residuals. mRNA half-lives data was taken from Schwanhäusser et al. (<https://www.nature.com/articles/nature10098>) and compared with the residuals calculated on the same genes from the mouse small intestine washes and tissues. We included genes with half-lives below 5 hours because Halpern et al, 2023, showed that small intestine shed cells half-life is around 1.5h (Figure 1g at Bahar Halpern et al, 2023).

2. Sample variability and reproducibility: The turnover scoring approach relies on samples obtained during surgical procedures or colonoscopies. Variability in patient conditions (e.g. surgical stress, diet) or technical handling may affect RNA profiles. The authors should clarify whether technical replicates, patient-matched samples or independent cohorts have been evaluated to assess the robustness of turnover scores (by revisiting their published data).

This is an important point, particularly relevant to the new human stomach wash shed cell data presented in our study. We have now added new experimental data that presents patient-matched repeats of 3 patients in which NGT fluids were sampled twice in 10-15 minutes intervals. We added one sample of control patient that we already have and added samples from two additional patients with upper GI tract malignancies (thus we didn't include them in main figure 3). We find that these biological repeats strongly cluster together (EV2A).

EV2A - Clustergram of NGT samples shows reproducibility across patients. Every column represents a sample and every row represents a gene. Colors denote biological repeats obtained at 10-15 minutes intervals.

We also discuss this important point as a potential limitation on the discussion, rows 255-257:

“Our method has several potential limitations. It relies on the mRNA half-life of shed cells, which can vary with physiological states such as stress, diet, or disease. Like other RNA-based techniques, it may be sensitive to technical factors, from sample collection to processing.”

3. Previous attempts to model cellular dynamics from scRNA-seq data: The current study focuses on using static transcriptomic snapshots to infer dynamic properties such as turnover with a very straightforward approach. Previous studies have similarly aimed to reconstruct cellular dynamics from scRNA-seq data using computational models, including those based on trajectory inference, barcoding or deterministic modelling frameworks. It would strengthen the manuscript to acknowledge these previous efforts and clearly position the current approach in relation to them.

We now discuss these important computational and barcoding approaches for reconstructing cellular dynamics from static single cell atlas snapshots in our introduction:

“Quantifying turnover has traditionally required pulse-chase methods, in which specific sub-populations are labeled(Leblond & Walker, 1956), either through DNA-integrating reagents (Bonhoeffer *et al*, 2000) or, more recently via cellular barcodes (Urbanus *et al*, 2023; Sankaran *et al*, 2022)”

And:

“Computational tools have been developed to infer cellular dynamics based on single cell RNAseq (scRNAseq) data through trajectory inferences (La Manno *et al*, 2018; Trapnell *et al*, 2014; Setty *et al*, 2019; Weiler *et al*, 2024), however, they do not directly inform on cellular turnover rates.”

4. Generalisability to disease contexts:

The authors mention potential applications in diseases such as IBD and cancer. However, disease-associated shedding events (e.g. necrosis, immune-mediated epithelial injury) may be fundamentally different from physiological turnover. This could affect the interpretation of turnover scores. The authors should briefly discuss limitations and necessary considerations when applying the method to diseased tissues, particularly in inflammatory or neoplastic contexts.

This is an important point that we now discuss. Indeed, we used a global reference for the upper GI tract tissue expression, whereas ideally biopsy per individual would be more precise (although technically limiting). We discuss this on rows 257-260:

“Disease-related shedding—triggered by conditions like neoplasia, necrosis, or inflammation—can alter the quantity and composition of shed material. In such cases, tailored sampling of both tissue and washes may improve accuracy.”

5. Availability of code and reproducibility:

While the authors mention that the code is available on request, and the mathematical formalism is rather straight-forward to implement, providing public access (e.g. via a publicly available repository such as GitHub) to a well-documented pipeline would greatly improve data and method sharing according to the FAIR principles.

We have now deposited all data and code on GITHUB (<https://github.com/barkait/TurnoverScore/>), Zenodo (<https://zenodo.org/records/15706697>) and in the Gene Expression Omnibus (GEO) under accession number GSE301268 (reviewer accession code mdsvkkimdvcbfqr), raw counts and metadata of NGT samples from figure 3 also appear on Table EV2 and EV3, respectively.

Minor points

- The Introduction would benefit from a brief reference to previous studies describing gene expression differences in apoptotic or shed epithelial cells. We have now added a reference to gene expression changes in shed cells in rows 42-44:

“While shed cells can experience changes in gene expression due to apoptosis or other death-related processes(Ngo *et al*, 2022), they retain the identity of the tissue and cell type of origin”

- The authors should provide more information on sample sizes and different scales of observed effects between samples/tissues (fold-changes, absolutes of turnover score)

We now added the number of cells/pixels for each boxplot (2E, 3EFG, 4E).

Reviewer #2:

This article 'Transcriptomic profiling of shed cells enables spatial mapping of cellular turnover in human organs' by Barkai et al. describes how cell turnover rates can be determined by comparing RNA expression profile of single cells in the gastrointestinal tract using bulk sequencing data generated from shed cells. The authors built on previous work from their lab where they elegantly demonstrated that shed cells can be used for bulk and single cell RNA transcriptomic read-outs in order to determine epithelial cell states. Extending this approach to determine cellular turnover rates would allow identification of aberrant turnover rates in specific gastrointestinal lineages in pathological contexts such as IBD and cancer, and could hold diagnostic value. This reviewer has the following questions / concerns that would need to be addressed:

The authors describe cellular turnover (here defined as 'the time between a cell's birth and death') using the difference between RNA levels in shed cells versus cells in tissue context. However, one might argue that there are additional factors influencing 'turnover potential' measured in this way, These include the loss of any cellular state due to differentiation into another state, differential release of RNA from different cell states and preferential apical versus basolateral shedding of particular lineages (e.g. Paneth cells may bias for basolateral clearance, all submucosal lineages in this study are not expected to lose most of their RNA to the intestinal lumen).

The authors should discuss these points. This can be partly experimentally addressed by performing single cell RNA/nuclei sequencing on shed cells directly allowing the comparison of cell type ratios.

We thank the reviewer for these important points. We now discuss all of these in the limitations section of our discussion on rows 260-264:

“Biological factors add complexity: differentiation may reduce representation of certain cell states, and RNA release not associated with cellular shedding may also occur. Directionality of shedding (apical vs. basolateral) introduces lineage-specific biases – for example, limiting the detection of submucosal lineages in luminal samples. Thus, the method is particularly informative for epithelial lineages.”

We have now also performed two new analyses – one comparing representation of shed tissue cells from the organs we focused on in serum (cell free RNA, cfRNA). We show that the representation of transcripts from the epithelial cells in the GI tract is much higher in the luminal washes compared to cfRNA (Figure EV4A). We also demonstrate that residuals in cfRNA are anti-correlated with those in washes, indicating that shedding into the blood stream is inversely correlated (Figure EV4C). It is indeed very probable that turnover of stromal cell types would be more highly represented in cfRNA, and we now explicitly discuss

this and explain that our methodology is particularly suited to epithelial cells rows 178-183:

“In addition, we found that overall colonic representation is more prominent in biopsy and wash compared to serum cfRNA (EV 4AB) and that biopsy-wash residuals are anti-correlated with biopsy-serum residuals indicating that shedding into the bloodstream is inversely zoned (EV 4C). Examination of colonic specific gene expression in both serum cfRNA and wash transcriptomics show that mature colonocytes and goblet cells have more representation in the luminal wash compared to the serum (EV 4D).”

And on row 264:

“Thus, the method is particularly informative for epithelial lineages.”

Although this may be beyond the scope of the study, an extensive shed single cell dataset may allow for more granular statements on cell turnover rates (e.g. for rare lineages like Tuft cells, M cells, enteroendocrine cell subtypes).

We have attempted to obtain shed single cell data from human gastric washes, but unfortunately this has been unfeasible due to the low viability percentage of the cells in nasogastric tube samples. We have now calculated turnover scores for more cell types using single shed cell atlas that we have previously reconstructed (EV1E).

Minor points

Some genes display somewhat unexpected residual patterns. *Lgr5* (stem cell gene) and *Defa22* (Paneth cell gene) are barely depleted in the shed fraction (Figure 2A), while *LCT* (enterocyte gene) surprisingly is among the depleted ones for the human duodenum (Figure 3D): Could the authors comment on this, could this deviation from the expected patterns e.g. correlated to expression levels?

The figures are at log scale and therefore might be misleading regarding the effect sizes. *Lgr5* and *Defa22* are depleted in the shed cell fraction at a fold-change of 0.57 and 0.48 respectively. Regarding the depletion of *LCT* in Figure 3D, indeed this is somewhat surprising, it could be related to particularly low RNA stability of this gene or other factors related to processes occurring after cell shedding. We now comprehensively discuss these and other potential limitations of our approach in the Discussion on page 10:

“Our method has several potential limitations. It relies on the mRNA half-life of shed cells, which can vary with physiological states such as stress, diet, or disease. Like other RNA-based techniques, it may be sensitive to technical factors, from sample collection to processing. Disease-related shedding – triggered by conditions like neoplasia, necrosis, or inflammation – can alter the quantity and composition of the shed cells or cell fragments. In such cases, tailored sampling of both tissue and washes may improve accuracy. Biological

factors add complexity: differentiation may reduce representation of certain cell states, and RNA release not associated with cellular shedding may also occur. Directionality of shedding (apical vs. basolateral) introduces lineage-specific biases – for example, limiting the detection of submucosal lineages in luminal samples. Thus, the method is particularly informative for epithelial lineages. Changes in gene expression after shedding (Bahar Halpern *et al*, 2023) could be taken into consideration in future developments of our approach.”

In line with the previous point: Stem cells are predicted to have higher turnover scores than intestinal goblet cells (Figure 3G), which is not expected.

We agree with the reviewer, this may be related to the low numbers of goblet cells in the data set analyzed (n=7), potentially capturing goblet cells from crypt/lower villus zones, thus showing low turnover scores.

In Figure 5, the authors identify a population of colonocytes that harbors an increased turnover rate. Could these gene sets also be induced after shedding already happened (e.g. through exposed TLR receptors)? A single cell comparison between shed cells and colonocytes could reveal this, if post-shedding signatures are induced it is expected that unique signatures at a single cell level would appear

We thank the reviewer for the suggestion, but unfortunately we were not able to capture enough viable shed colonocytes at a sufficient amount and quality for single cell RNA sequencing experiment.

Reviewer #3:

In their manuscript, Barkai et al. aim to estimate cellular turnover rates at single-cell resolution by integrating bulk RNA-seq from tissue and shed cells, along with both single-cell and spatial transcriptomic data. Turnover rates in the human esophagus, stomach and small intestine are inferred based on RNA sequencing on shed cells from the upper gastrointestinal tract, in addition to inference of turnover potential in the human large intestine based on colonic fecal washes. The study provides a map of cellular turnover rates across tissues, cell types, and tissue microenvironments that potentially promote cell death. One intriguing finding is a subset of short-lived, interferon-stimulated colonocytes within a pro-inflammatory microenvironment.

This work advances our ability to better understand and characterize turnover rates in cellular populations, in a comparative manner, which is an important and timely challenge, with implications for healthy and pathological tissue states. The data generated in this study could also be used as a valuable resource for future studies. A few comments are elaborated below.

Comments:

- In the first section of the results, the process by which turnover scores are computed should be explained more clearly (consider adding relevant

equations?). Since this is the basis for the entire study, it is crucial to clarify this point.

We have now added equation and explanations:

“Equation 1, where $Residual_i$ is the residual of $gene_i$ and $Expression_i$ is the sum normalized expression of $gene_i$ ”

- Have you tried assigning turnover potential scores per cell directly at the cellular level and not via the gene-based analysis (which has its limitations, for example due to the assumption of linear combination of residuals).

Our turnover score indeed provides a score per cell, rather than per gene. We integrate the discordance in expression of all genes between the tissue and shed cell fractions (Equation (1)). We indeed also show scatter plots of the individual gene expression between the shed cell fraction and the tissue fraction, we now clarify in each panel whether the data is gene-based or cell-based.

- The Discussion section should also include discussion of limitations of the study, including the linearity assumption in the calculation of turnover scores, microenvironmental signatures showing correlation and not causal effects (without additional perturbations for example), the fact that the whole calculation of turnover scores is done by regressing over bulk measurements which may include convolution of signals coming from different cells, etc.

We now discuss these important points on rows 267-269:

“Methodologically, turnover scores are derived from linear regression on bulk RNA data, which integrates signals from mixed populations. This can limit resolution of cell-type-specific dynamics. Moreover, microenvironmental signatures may reflect correlations rather than causality.”

- It's better to have the code publicly available (e.g. on GitHub) and not only make it available upon request.

We have now included all code in GitHub

(<https://github.com/barkait/TurnoverScore>) and Zenodo:

<https://zenodo.org/records/15706697>

- There were several references to Methods sections from the main text which do not seem to correspond to any of the existing methods sections (e.g. in the Results subsection 'Spatial signatures of high turnover epithelial cells').

We have now corrected this and specifically refer to the External datasets section in the Methods.

- Are there single-cell measurements of shed cells? If so, would it have been possible to map these directly to the spatial data or single-cell data to identify potential for shedding?

We now compare our results and estimates of zonal shedding in the mouse small intestine with a single shed cell atlas that we have previously reconstructed,

demonstrating that the shed-cells turnover score is similar to enterocytes at the very tip of the villus (V6 zone, EV1C).

- Typo: 'Our approach therefore expose
We have now corrected this typo.

18th Aug 2025

Manuscript Number: MSB-2025-12964R

Title: Transcriptomic profiling of shed cells enables spatial mapping of cellular turnover in human organs

Dear Prof Itzkovitz,

Thank you for the submission of your revised manuscript to Molecular Systems Biology. We have now received the enclosed reports from the referees that were asked to re-assess it. As you will see the reviewers are now globally supportive and I am pleased to inform you that we will be able to accept your manuscript pending the following final amendments:

1) Please provide updated contact email addresses for the following authors since their current emails bounced: Keren Bahar Halpern - kerenb@weizmann.ac.il and Tal Barkai - tal.barkai@weizmann.ac.il

2) In the main manuscript file, please include keywords to max. 5.

3) Please rename the "Data and code availability" section to "Data availability" and format according to the example below. Please also note that publicly available datasets that have been analyzed should not be included in the Data Availability statement, but rather should be referenced in the Methods (please also see our suggestion to include these as Data Citations below):

"The datasets and computer code produced in this study are available in the following databases:

- Chip-Seq data: Gene Expression Omnibus GSE46748 (<https://www.ncbi.nlm.nih.gov/geo/query/acc.cgi?acc=GSE46748>)

- Modeling computer scripts: GitHub (<https://github.com/SysBioChalmers/GECKO/releases/tag/v1.0>)

- [data type]: [full name of the resource] [accession number/identifier] ([doi or URL or identifiers.org/DATABASE:ACCESSION])"

4) As you have indicated that the raw sequencing data is not available due to ethical restrictions and patient confidentiality, if the data might be available with controlled access, please include the following in the Data Availability statement: precise conditions of access (including contact details for access requests), a timeframe for response to requests and details of any restrictions imposed on data use via data use agreements.

5) Please release the data in GEO accession GSE301268 so that it is publicly available (and remove the reviewer accession code information from the Data Availability statement).

6) Our journal encourages inclusion of *data citations in the reference list* to directly cite datasets that were re-used and obtained from public databases. Data citations in the article text are distinct from normal bibliographical citations and should directly link to the database records from which the data can be accessed. In the main text, data citations are formatted as follows: "Data ref: Smith et al, 2001" or "Data ref: NCBI Sequence Read Archive PRJNA342805, 2017". In the Reference list, data citations must be labeled with "[DATASET]". A data reference must provide the database name, accession number/identifiers and a resolvable link to the landing page from which the data can be accessed at the end of the reference. Further instructions are available at .

7) Please rename "Competing Interests" to "Disclosure and competing interests statement". We updated our journal's competing interests policy in January 2022 and request authors to consider both actual and perceived competing interests. Please review the policy <https://www.embopress.org/competing-interests> and update your competing interests if necessary.

8) Please remove the author contributions from the manuscript and specify author contributions in our submission system. CRedit has replaced the traditional author contributions section because it offers a systematic machine-readable author contributions format that allows for more effective research assessment. You are encouraged to use the free text boxes beneath each contributing author's name to add specific details on the author's contribution. More information is available in our guide to authors:

<https://www.embopress.org/page/journal/17574684/authorguide#authorshipguidelines>

9) In the Methods, please take care of the following:

- The Materials and Methods section should be renamed to "Methods".

- The use of human samples requires information on the authority granting ethics approval (e.g. IRB) and informed consent. If the need for approval is waived, please cite the reason (e.g. non-human subject research because the samples used were de-identified/coded with no identifying information) and legislation in the relevant methods section.

- If approval was required, please ensure that the name of the body that provided the authorization is included in the Methods.

- Please also state that the experiments conformed to the principles set out in the WMA Declaration of Helsinki and the Department of Health and Human Services Belmont Report. Please note that this is a separate statement from the specific ethics committee approval and informed consent.

- You have indicated in the Reagents and Tools table that primers have been used in your study. Please ensure primer sequences used are included in the Methods, and that the Author Checklist is updated with this information.

- Please ensure that a statement on whether or not blinding was done is included in the Methods even if no blinding was done. Please also be sure to update the Author Checklist with this information and where it can be found in the manuscript.

- When submitting your revised manuscript, please do not include the Reagents and Tools Table in the Methods section of the manuscript but only upload it as a separate file choosing the file type "Reagent Table".
- 10) Please remove the section on "Lead contact" from the manuscript.
- 11) Please place individual sections of the manuscript in the following order: Title page - Abstract & Keywords - Introduction - Results - Discussion - Methods - Data Availability - Acknowledgements - Disclosure and Competing Interests Statement - References - Figure Legends - Expanded View Figure Legends.
- 12) For the figures and figure legends, please take care of the following:
 - Please remove all figures from main manuscript file and leave only main and EV figure legends placed after the references.
 - Please make sure to update the callouts of all figures in the main manuscript text. Currently figure callouts are missing for Figure EV4. In addition, please replace the word "supplementary" with "Expanded View" in the Data Availability statement.
 - Please note that the box plots need to be defined in terms of centre and percentile in the legends of figures 2E, 3E-G; 4E, EV1 C-E; EV4 A, B
 - Please note that information related to n is missing in the legends of figure EV4 D
- 13) For EV Tables, please remove the table legends from the main manuscript file to above the table itself. Tables EV2-EV4 look like datasets and should be uploaded and renamed as Dataset EV1-EV3. The nomenclature should be updated in all places (source file names, titles in the submission system, legends, manuscript callouts). For EV Datasets, the legend should also be removed from the manuscript but it should be added to the corresponding file in a separate tab.
- 14) Please remove the "Supplemental Information" section with titles/legends from the manuscript, as these should be included elsewhere as requested above.
- 15) Please ensure that all funding sources are entered into the manuscript submission system (i.e. please add xx). Currently only one funder has been entered into our submission system, the following also need to be added: the Moross Integrated Cancer Center, the Helen and Martin Kimmel Award for Innovative Investigation, the Yad Abraham Research Center for Cancer Diagnostics and Therapy, the Israel Science Foundation grants no. 908/21 and no. 3663/21; Weizmann-Sheba joint research grant and a research grant from the Ministry of Innovation, Science and Technology, Israel.
- 16) Please provide a synopsis image that summarises the main findings of the manuscript on a glance and upload it as a high-resolution jpeg file 550 pixels wide x (300-600) pixels high.
- 17) Please clarify the source(s) of the microscopy images in the manuscript. Currently it is unclear which paper and specific figure these panels originate from. We would suggest making this clearer in the legend of the each figure in which images have been reused.
- 18) As part of the EMBO Publications transparent editorial process initiative (see our policy here: https://www.embopress.org/transparent-process#Review_Process), Molecular Systems Biology will publish online a Peer Review File (PRF) to accompany accepted manuscripts. This file will be published in conjunction with your paper and will include the anonymous referee reports, your point-by-point response and all pertinent correspondence relating to the manuscript. Let us know whether you agree with the publication of the PRF and as here, if you want to remove or not any figures from it prior to publication. Please note that the Authors checklist will be published at the end of the PRF.
- 19) After your paper is published, we may promote it on social media. If you have any handles or hashtags for Bluesky you would like included, please let us know.
- 20) Please provide a point-by-point letter INCLUDING my comments and your detailed responses (as Word file).

I look forward to reading a new revised version of your manuscript as soon as possible.

Yours sincerely,

Poonam Bheda, PhD
Scientific Editor
Molecular Systems Biology

Reviewer #1:

The authors have thoroughly and appropriately addressed all my (and other) major and minor concerns raised during review. The key improvements include:

- 1.) Validation of robustness: The new analyses excluding apoptotic genes and evaluating mRNA half-life effects substantially strengthen the methodological foundation in my eyes.
- 2.) Enhanced reproducibility: The addition of patient-matched biological replicates provides an impression of the overall coherence within the data.
- 3.) Improved contextualisation: The manuscript now properly situates the work within the broader field of cellular turnover inference.

4.) Comprehensive limitations: The expanded discussion appropriately acknowledges potential confounders and boundaries of the applicability.

5.) Open science compliance: Full code and data availability enables reproducibility and future extensions, which is very important and is highly appreciated!

The minor points regarding sample sizes, gene expression changes in shed cells, and figure clarifications have also been adequately addressed. The manuscript now represents a significant conceptual and methodological advance that will be of broad interest to the systems biology, gastroenterology, and computational biology communities.

Reviewer #2:

The authors have satisfactorily addressed my comments, and I would like to congratulate them on conducting such a compelling and well-executed study.

Reviewer #3:

The authors have satisfactorily addressed all of my comments and suggestions.

1) Please provide updated contact email addresses for the following authors since their current emails bounced: Keren Bahar Halpern - kerenb@weizmann.ac.il and Tal Barkai - tal.barkai@weizmann.ac.il

Copied from WIS outlook: kerenb@weizmann.ac.il tal.barkai@weizmann.ac.il

2) In the main manuscript file, please include keywords to max. 5.

We added keywords after the abstract section.

3) Please rename the "Data and code availability" section to "Data availability" and format according to the example below. Please also note that publicly available datasets that have been analyzed should not be included in the Data Availability statement, but rather should be referenced in the Methods (please also see our suggestion to include these as Data Citations below):

"The datasets and computer code produced in this study are available in the following databases:

- Chip-Seq data: Gene Expression Omnibus GSE46748

(<https://www.ncbi.nlm.nih.gov/geo/query/acc.cgi?acc=GSE46748>)

- Modeling computer scripts: GitHub

(<https://github.com/SysBioChalmers/GECKO/releases/tag/v1.0>)

- [data type]: [full name of the resource] [accession number/identifier] ([doi or URL or identifiers.org/DATABASE:ACCESSION])"

Renamed and the data availability section was changed to the format as above. List of publicly available dataset is now under "External datasets" section of "Methods".

4) As you have indicated that the raw sequencing data is not available due to ethical restrictions and patient confidentiality, if the data might be available with controlled access, please include the following in the Data Availability statement: precise conditions of access (including contact details for access requests), a timeframe for response to requests and details of any restrictions imposed on data use via data use agreements.

The data is not available even with controlled access.

5) Please release the data in GEO accession GSE301268 so that it is publicly available (and remove the reviewer accession code information from the Data Availability statement).

The data is publicly available and the accession code information has been removed from the Data Availability statement.

6) Our journal encourages inclusion of *data citations in the reference list* to directly cite datasets that were re-used and obtained from public databases. Data citations in the article text are distinct from normal bibliographical citations and should directly link to the database records from which the data can be accessed. In the main text, data citations are formatted as follows: "Data ref: Smith et al, 2001" or "Data ref: NCBI Sequence Read Archive PRJNA342805, 2017". In the Reference list, data citations must be labeled with "[DATASET]". A data reference must provide the database name, accession number/identifiers and a resolvable link to the landing page from which the data can be accessed at the end of the reference. Further instructions are available at <https://www.embopress.org/page/journal/17574684/authorguide#referencesformat>.

7) Please rename "Competing Interests" to "Disclosure and competing interests statement". We updated our journal's competing interests policy in January 2022 and request authors to consider both actual and perceived competing interests. Please review the policy <https://www.embopress.org/competing-interests> and update your competing interests if necessary.

Renamed.

8) Please remove the author contributions from the manuscript and specify author contributions in our submission system. CRediT has replaced the traditional author contributions section because it offers a systematic machine-readable author contributions format that allows for more effective research assessment. You are encouraged to use the free text boxes beneath each contributing author's name to add specific details on the author's contribution. More information is available in our guide to authors:

<https://www.embopress.org/page/journal/17574684/authorguide#authorshipguidelines>

Removed.

9) In the Methods, please take care of the following:

- The Materials and Methods section should be renamed to "Methods". Renamed
- The use of human samples requires information on the authority granting ethics approval (e.g. IRB) and informed consent. If the need for approval is waived, please cite the reason (e.g. non-human subject research because the samples used were de-identified/coded with no identifying information) and legislation in the relevant methods

section. This is detailed under “Upper GI fluid collection” section: Helsinki #SMC-8665-21

- If approval was required, please ensure that the name of the body that provided the authorization is included in the Methods. This is detailed under “Upper GI fluid collection” section: Helsinki #SMC-8665-21

- Please also state that the experiments conformed to the principles set out in the WMA Declaration of Helsinki and the Department of Health and Human Services Belmont Report. Please note that this is a separate statement from the specific ethics committee approval and informed consent. Added.

- You have indicated in the Reagents and Tools table that primers have been used in your study. Please ensure primer sequences used are included in the Methods, and that the Author Checklist is updated with this information. Primers were added to the methods section an Author Checklist has been updated.

- Please ensure that a statement on whether or not blinding was done is included in the Methods even if no blinding was done. Please also be sure to update the Author Checklist with this information and where it can be found in the manuscript.

“Patients arriving for a medical procedure were recruited without blinding at the general surgery department of Sheba Medical Center.” Author checklist was updated.

- When submitting your revised manuscript, please do not include the Reagents and Tools Table in the Methods section of the manuscript but only upload it as a separate file choosing the file type "Reagent Table". Table was removed from the main text.

10) Please remove the section on "Lead contact" from the manuscript. Removed.

11) Please place individual sections of the manuscript in the following order: Title page - Abstract & Keywords - Introduction - Results - Discussion - Methods - Data Availability - Acknowledgements - Disclosure and Competing Interests Statement - References - Figure Legends - Expanded View Figure Legends.

Done.

12) For the figures and figure legends, please take care of the following:

- Please remove all figures from main manuscript file and leave only main and EV figure legends placed after the references. Done.

- Please make sure to update the callouts of all figures in the main manuscript text. Currently figure callouts are missing for Figure EV4. In addition, please replace the word "supplementary" with "Expanded View" in the Data Availability statement. Added callout

for Figure EV4.

- Please note that the box plots need to be defined in terms of centre and percentile in the legends of figures 2E, 3E-G; 4E, EV1 C-E; EV4 A, B **Boxplots were defined.**

- Please note that information related to n is missing in the legends of figure EV4 D **Number of genes per cell type were added to the figure legend.**

13) For EV Tables, please remove the table legends from the main manuscript file to above the table itself. Tables EV2-EV4 look like datasets and should be uploaded and renamed as Dataset EV1-EV3. The nomenclature should be updated in all places (source file names, titles in the submission system, legends, manuscript callouts). For EV Datasets, the legend should also be removed from the manuscript but it should be added to the corresponding file in a separate tab.

Done.

14) Please remove the "Supplemental Information" section with titles/legends from the manuscript, as these should be included elsewhere as requested above. **Removed.**

15) Please ensure that all funding sources are entered into the manuscript submission system (i.e. please add xx). Currently only one funder has been entered into our submission system, the following also need to be added: the Moross Integrated Cancer Center, the Helen and Martin Kimmel Award for Innovative Investigation, the Yad Abraham Research Center for Cancer Diagnostics and Therapy, the Israel Science Foundation grants no. 908/21 and no. 3663/21; Weizmann-Sheba joint research grant and a research grant from the Ministry of Innovation, Science and Technology, Israel. **We have ensured this.**

16) Please provide a synopsis image that summarises the main findings of the manuscript on a glance and upload it as a high-resolution jpeg file 550 pixels wide x (300-600) pixels high.

Provided.

17) Please clarify the source(s) of the microscopy images in the manuscript. Currently it is unclear which paper and specific figure these panels originate from. We would suggest making this clearer in the legend of the each figure in which images have been reused.

We now clarified in the figure legends of Figure 4 and Figure 5 that the microscopy and the gene expression data are from Oliver, 2025.

18) As part of the EMBO Publications transparent editorial process initiative (see our policy here: https://www.embopress.org/transparent-process#Review_Process), Molecular Systems Biology will publish online a Peer Review File (PRF) to accompany accepted manuscripts. This file will be published in conjunction with your paper and will include the anonymous referee reports, your point-by-point response and all pertinent correspondence relating to the manuscript. Let us know whether you agree with the publication of the PRF and as here, if you want to remove or not any figures from it prior to publication. Please note that the Authors checklist will be published at the end of the PRF.

We agree to publish the PRF as is.

19) After your paper is published, we may promote it on social media. If you have any handles or hashtags for Bluesky you would like included, please let us know.

We don't have any.

20) Please provide a point-by-point letter INCLUDING my comments and your detailed responses (as Word file).

Please note - We also added technical details in the methods section (e.g. pseudo number, etc.).

18th Sep 2025

Manuscript number: MSB-2025-12964RR

Title: Transcriptomic profiling of shed cells enables spatial mapping of cellular turnover in human organs

Dear Prof Itzkovitz,

Congratulations on an excellent manuscript, I am pleased to inform you that your manuscript has been accepted for publication in Molecular Systems Biology. Thank you for your comprehensive response to referee concerns. It has been a pleasure to work with you to get this to the acceptance stage.

Yours sincerely,

Sincerely,

Poonam Bheda, PhD
Scientific Editor
Molecular Systems Biology
